# FROM TOKENS TO WORDS:
# ON THE INNER LEXICON OF LLMS

**Guy Kaplan, Matanel Oren, Yuval Reif, and Roy Schwartz**
The Hebrew University of Jerusalem
{guy.kaplan3,matanel.oren,yuval.reif,roy.schwartz1}@mail.huji.ac.il

## ABSTRACT

Natural language is composed of words, but modern large language models (LLMs) process *sub-words* as input. A natural question raised by this discrepancy is whether LLMs encode words internally, and if so how. We present evidence that LLMs engage in an intrinsic detokenization process, where subword sequences are combined into coherent whole-word representations at their last token. Our experiments show that this process primarily takes place within the early and middle layers of the model. We further demonstrate its robustness to arbitrary splits (e.g., "cats" to "ca" and "ts"), typos, and importantly—to out-of-vocabulary words: when feeding the last token internal representations of such words to the model as input, it can "understand" them as the complete word despite never seeing such representations as input during training. Our findings suggest that LLMs maintain a latent vocabulary beyond the tokenizer's scope. These insights provide a practical, finetuning-free application for expanding the vocabulary of pre-trained models. By enabling the addition of new vocabulary words, we reduce input length and inference iterations, which reduces both space and model latency, with little to no loss in model accuracy.[1]

## 1 INTRODUCTION

Large language models (LLMs) rely heavily on tokenization methods such as byte-pair encoding (BPE; Sennrich et al., 2016). Such methods often split words into multiple tokens, potentially disrupting their morphological structure (Batsuren et al., 2024).[2] Typos and other perturbations can also lead to large variations in the tokens that represent a word (Kaushal & Mahowald, 2022). Nonetheless, LLMs exhibit a remarkable ability to recover word meaning (Cao et al., 2023). This raises important questions about how models internally compose meaningful word representations from tokens, a process referred to as *detokenization* (Elhage et al., 2022; Gurnee et al., 2023).

In this work, we seek to understand the detokenization mechanism in LLMs. We consider two cases: words that are not part of the model's BPE vocabulary, and are thus split into multiple sub-words; and single-token, in-vocabulary words that we artificially split into multiple tokens. In both cases, our experiments indicate a word-level detokenization process in LLMs, which occurs mainly in the early to middle layers. Our results hint that language models hold a *latent vocabulary* or *inner lexicon*, which they access to identify words from token sequences.[3]

We begin by examining if internal representations of token sequences reflect whether or not a sequence of tokens comprises a word (Sec. 3). We probe the model's hidden representations (Conneau et al., 2018) of both multi-token real English words and gibberish *nonwords* (Frisch et al., 2000). We observe that the representations of words and nonwords substantially diverge in middle layers—a simple k-nearest neighbors classifier achieves a 89% accuracy on the task of discriminating between the two groups. Overall, these results suggest that models hold a concept of recognized words.

We next explore the mechanism through which models reconstruct cohesive word representations from sub-word tokens (Sec. 4). To do so, we use techniques that interpret a token's hidden states and

---

[1]We release our code at `https://github.com/schwartz-lab-NLP/Tokens2Words`.

[2]For instance, the word "unhappiness" might be tokenized into "un," "h," and "appiness" (see Fig. 1)

[3]This process might resemble the *mental lexicon* in humans (Aitchison, 2012; Marslen-Wilson et al., 1994).

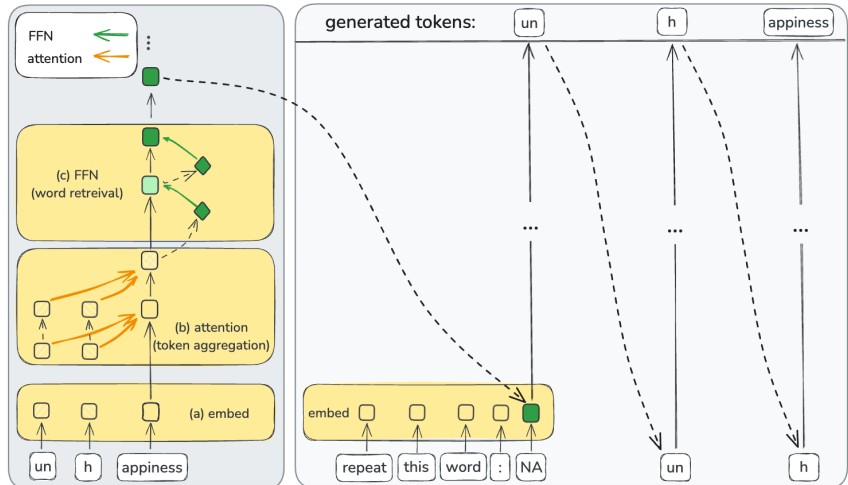

Figure 1: **Left**: The sub-word detokenization process in LLMs. From bottom to top: **(a) Tokenization and Embedding**: The input string is tokenized using a sub-word tokenizer (e.g., BPE) and converted into vector embeddings; **(b) Token Aggregation**: The attention mechanism relays information from the word's previous tokens ("un", "h") into its final sub-word representation ("appiness"); **(c) Word Retrieval**: The model retrieves the full word representation from an implicit internal lexicon in its feedforward (FFN) layers. This representation is added to the residual stream, until it takes over the word's hidden representation.

**Right**: When taking this hidden representation and patching it into another prompt, the model interprets it as the original word. In this example, the model is prompted to repeat the (single vector) hidden representation of the word "unhappiness" (originally represented as three tokens), and is able to "understand" it by regenerating the original three tokens.

decode them into natural language (Belrose et al., 2023; Ghandeharioun et al., 2024). We find that for both multi-token and single-token words, in most cases, the last token can be decoded as the full word after being processed by 3–5 layers, with some words requiring up to 15 layers. Interestingly, 23% of the multi-token words are never successfully decoded from their last token's hidden states, hinting this inner lexicon does not cover all words.

We then turn to explore how models assemble these full word representations (Sec. 5). We first interpret the feedforward network (FFN) updates in the vocabulary space (Geva et al., 2021; Todd et al., 2024). We show that in 85% of the evaluated words, an FFN update promoting the full word's concept is written to the last token's residual stream (Geva et al., 2022; Merullo et al., 2024). Importantly, this happens in the layers *before* the full word's representation emerges, indicating that models retrieve the reconstructed word representation from their FFN weights. Second, we compare attention patterns between standard single-token words to how multi-token words attend to their preceding sub-word tokens. We find that the last token in multi-token words attends significantly *more* to its previous tokens (which are sub-words of the same word) in layers 1–2 compared to single-token words (where these previous tokens represent other words), and significantly *less* in the following layers. This suggests that models initially aggregate information from previous tokens, and later almost ignore them. See Fig. 1 for an overview.

Our findings have concrete applications (Sec. 6); they allow us to expand the model input and output BPE vocabulary with the fused representation of multi-token words found in our experiments. Importantly, this expansion requires no update to model parameters. Applying our approach to multi-token words found in English Wikipedia data (Merity et al., 2017), we find that the model successfully uses the new vocabulary entries both as inputs and during generation: its language model performance is maintained, and even slightly improves. This demonstrates the potential to dramatically reduce both input and output sequence length, and inference costs accordingly, especially in languages where the ratio of tokens per word is high (Ahia et al., 2023; Petrov et al., 2023).

Overall, our results establish that word-level detokenization is a core process in LLMs, and provide evidence of how it unfolds across model layers. Beyond improving our understanding of the internal

mechanisms driving LLMs, our work lays the foundation for practical applications, particularly in optimizing token management and reducing computational costs.

## 2 RELATED WORK

**Tokenization**   Sub-word tokenization algorithms (Wu, 2016; Kudo, 2018) are the standard for preprocessing text in modern LLMs. The most widely used method is byte-pair encoding (BPE; Sennrich et al., 2016), which keeps frequent words intact and splits rare ones into multiple sub-words. Recent studies proposed ways to improve tokenization to consider word structure (Provilkov et al., 2020; Hofmann et al., 2022; Yehezkel & Pinter, 2023; Bauwens & Delobelle, 2024) or analyze how tokenization affects model performance (Bostrom & Durrett, 2020; Church, 2020; Klein & Tsarfaty, 2020; Zouhar et al., 2023; Schmidt et al., 2024). To the best of our knowledge, we are the first to thoroughly investigate how LLMs internally reconstruct word representations.

**Detokenization and stages of inference**   Early LLM layers have been shown to integrate local context and map raw token embeddings into representations of concepts or entities—a process called *detokenization*. However, such observations were based on specific case studies (Elhage et al., 2022; Lad et al., 2024). More generally, recent work showed early layers provide local syntactic information (Tenney et al., 2019; Vulić et al., 2020; Durrani et al., 2020; Sajjad et al., 2022) or focus on extracting information from previous tokens (Ben Artzy & Schwartz, 2024). Ferrando & Voita (2024) analyzed token attributions in LLMs, observing attention heads that promote sub-word merging (Correia et al., 2019). Our work focuses on word-level detokenization, and goes beyond previous efforts to provide an in-depth analysis of how word representations are assembled from multiple tokens.

**Interpreting the residual stream**   Recent methods for interpreting the intermediate states of LLMs draw on a *residual stream* perspective: the hidden state acts as an information stream along the layers, from which information is read at each layer, and new information is added through residual connections (Elhage et al., 2022). Thus, hidden states at any layer can be projected into the model's vocabulary space, treating the hidden state as if it were the output of the last layer (nostalgebraist, 2020; Dar et al., 2023; Belrose et al., 2023; Yom Din et al., 2024). Similarly, Ghandeharioun et al. (2024) proposed to decode information from hidden representations into natural language, by patching (Zhang & Nanda, 2024) it into a prompt that encourages the model to *verbalize* the encoded information. We use both approaches to inspect how token representations evolve across layers.

**LLM memories**   The idea of an *inner lexicon* aligns with recent work showing feedforward networks (FFN) layers in transformers act as key-value memories that encode factual and linguistic knowledge (Geva et al., 2021; Meng et al., 2022b; Dai et al., 2022). Particularly, FFNs were shown to enrich entity tokens with associated information (Meng et al., 2022b; Geva et al., 2023) and promote relevant concepts in vocabulary space to build up predictions (Geva et al., 2022). Our work expands on these findings, showing that word representations are pulled from FFN layers before emerging in the hidden state of the word's last token.

**Inner lexicon structure**   Inspired by studies on how concepts are encoded in LLMs (Park et al., 2024; 2025) and gradually promoted throughout their layers (Geva et al., 2022; Merullo et al., 2024), we consider the *inner lexicon* a "soft" lexicon, which (a) combines multiple vectors to form word representations (rather than a key-value dictionary); and (b) is not unique, i.e., a word might be stored and retrieved in more than one layer. Concurrently, Feucht et al. (2024) present evidence for an implicit vocabulary in LLMs, showing that models "forget" preceding tokens in multi-token words or multi-word entities, but remember previous tokens when processing single-token words.

## 3 A MOTIVATING OBSERVATION: LLMS CAN TELL WORDS FROM NON-WORDS

One of our key hypotheses in this work is that LLMs hold an internal lexicon of words, which is different from the BPE lexicon. We begin by asking whether LLMs, when processing a sequence of tokens, capture some notion of whether or not this sequence forms a word.

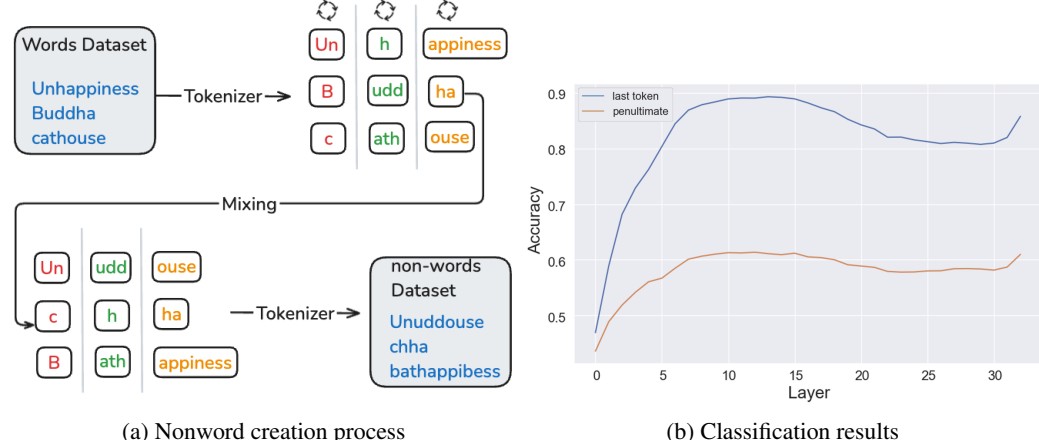

(a) Nonword creation process            (b) Classification results

Figure 2: Our word vs. nonword probing classification experiments. (2a) **Dataset creation process**. Top: words from the Gutenberg corpus are tokenized using the Llama2 tokenizer. Bottom: nonwords are generated by shuffling tokens while maintaining their positions within the word; (2b) **Classification results of words vs. nonwords**. Using the last token shows strong results (reaching up to 90% accuracy), which sharply rise in early layers (3–7), peak in the middle (13), and decrease in later layers (20–32). Using the penultimate token shows substantially lower scores, suggesting that the high classification accuracy is specifically tied to the presence of a complete word rather than token co-occurrence patterns.

To address this question, we construct a balanced dataset containing two groups: one with real English words, and another with artificially generated, meaningless *nonwords* (Frisch et al., 2000). Both groups are tokenized using the Llama2 tokenizer (Touvron et al., 2023). The word dataset consists of 10,000 distinct words sampled from the Gutenberg corpus (Gerlach & Font-Clos, 2018), with 53% of the words containing two tokens, 37.3% containing three tokens, and the rest four tokens. We generate the nonwords by shuffling tokens from the word dataset, ensuring that the prefix and suffix positions align with the original tokens' positional probabilities. For example, the token "ing", extracted from the final position of real words in the dataset (e.g., "directing" tokenized as "direct", "ing"), is retained as a suffix in the nonwords dataset. This process ensures that nonword tokens maintain position distribution properties similar to word tokens, preserving natural positional patterns and mitigating potential distributional biases (Fig. 2a). We next apply a $k$-nearest neighbors ($k$-NN) probing classifier ($k = 4$, using Euclidean distance) on the hidden states of the last tokens of both words and nonwords, for each layer of the Llama2-7B model. The training set consists of 80% of the dataset, and the remaining 20% are used for evaluation.

Our results (Fig. 2b, blue) reveal a three-stage pattern in the model's representation of word and nonword token sequences. In the model's first few layers, representations from both groups are relatively indistinguishable and accuracy is close to chance level. Then, from layers 2 to 6, a clear distinction between the two groups emerges, until the representations are almost completely separate in middle layers, between layers 6 and 20. At this point, the probe achieves a stable, high accuracy, peaking at 89% on layer 13. Finally, accuracy slightly drops after layer 20.[4]

Our results indicate the LLMs can distinguish between words and nonwords. But this distinction can be attributed to the distributional properties of words: it might be the case that models are recognizing common sequences of tokens, rather than identifying whole words. To test this hypothesis, we repeat the same experiment, this time using the *penultimate* token representation, for words three tokens or longer. By definition, such words also as frequently co-occur with the initial sub-word tokens in their words as the final tokens. Our results (Fig. 2b, orange line) show that the probe only reaches 61% classification accuracy, indicating that the high classification accuracy does not stem from token co-occurrence, but is tied to the presence of a whole word. See App. A for an analysis of misclassified words and nonwords.

---

[4]We perform a similar experiment using a dataset of morphologically plausible nonwords (ARC Nonword Database; Rastle et al. 2002), which shows a similar trend. See App. A.

Overall, our results show that language models internally represent words and nonwords differently. This distinction is gradually developed in the model's early layers and maintained throughout its middle layers. These findings support the hypothesis that the model performs a detokenization process and suggests where this process occurs. Building on these results, we next investigate how sub-word tokens are combined into word representations across model layers, and explore the internal mechanisms that facilitate this transformation.

# 4 EXTRACTING WORD IDENTITY FROM LLM HIDDEN STATES

We have so far observed that LLMs can differentiate between words and nonwords, suggesting an internal detokenization process specific to word composition. We next dig into this process, by asking whether we can directly extract word identity from the hidden states of sub-word tokens. We start by considering single-token words (Sec. 4.1), and then move on to multi-token words (Sec. 4.2).

## 4.1 SINGLE-TOKEN WORDS

We first consider in-vocabulary words, which are mapped to single tokens by the tokenizer. Naively, such words don't tell us much about detokenization, as they are represented using only one token. To address this, we artificially split them into multiple sub-word tokens. For example, we take the single-token word "cats" and split it into two tokens: "ca" and "ts". We hypothesize that if the model performs detokenization, it will represent the last token of the word ("ts") similarly to the original word token ("cats"). We iterate the WIKITEXT-103 dataset (Merity et al., 2017) and randomly split each single-token word longer than three characters into 2–5 sub-words tokens. We then feed the model the new sequence of tokens preceded by the last 100 tokens that came before each split word in the original text as context.[5]

To measure the similarity between the representation of the final token and original word, we apply the logit lens method (nostalgebraist, 2020), an interpretability method that maps a hidden representation of a given token to the word whose vector in the output unembedding matrix is most closely aligned with the hidden representation. We note that this technique is typically used to inspect model predictions of the *next* token in intermediate layers. As we aim to identify how LLMs represent the *current* word, we use the *input* embeddings matrix rather than the unembedding one.[6]

We study four LLMs: Llama2-7B, Llama3-8B (Dubey et al., 2024), Mistral-7B (Jiang et al., 2023), and Yi-6B (AI et al., 2024). For each layer, we report the rate of retrieval—the proportion of words for which the closest vector in the vocabulary space is the original (single token) word.

Our results for the Llama2-7B are shown in Fig. 3a.[7] Starting at layer 8, the hidden state of the final token is mapped to the original word with high accuracy, which peaks at more than 80% in layer 15. Interestingly, this accuracy then starts to decline, possibly because the model transitions into representing the prediction of the next token. We also find that 93.2% of all split words are correctly mapped to the original word in *at least one* model layer (see App. C.1).

Our results indicate that LLMs perform a detokenization process in cases where a single-token word is split into multiple tokens. This process assigns the final token with a hidden representation similar to that of the full, single-token word. But **how robust is this process**? To address this question, we consider a different way of splitting a word into sub-tokens—adding typos. For each word, we randomly flip two adjacent characters, delete a character, or insert a new character (see App. D for more details). Much like our previous experiment, this process splits the word into multiple tokens, as corrupted words are rarely found in BPE lexicons. We repeat the same experiment as above, aiming to study whether models map the last token to the original (correctly-spelled) word.

---

[5] We observe similar trends, in this and in other experiments in this section, when passing the split words *without* their context, albeit with slightly lower rates of retrieval.

[6] We also compute a second measure, cosine similarity, to compare between the hidden representations and input embeddings. We find similar results to those obtained using logit lens. For details, see App. F.

[7] Results for the other models (App. C) show a similar trend.

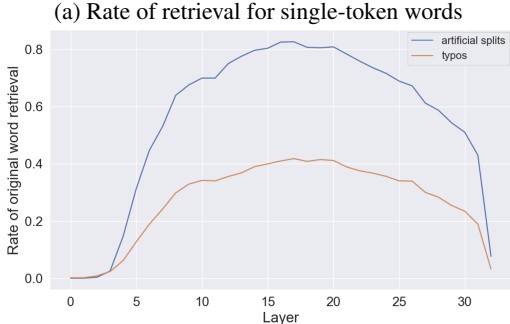
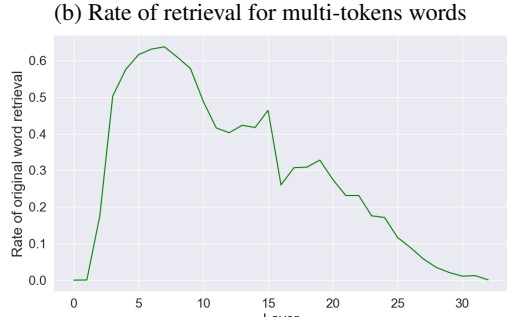

Figure 3: (3a) Logit lens rate of retrieval of single-token words artificially split (blue line) and split due to typos (orange). In both cases, we see an increasing rate of retrieval after the 4th layer, peaking in the middle (16–17 layers) and then dropping; (3b) Patchscopes rate of retrieval for multi-token words. In this case results peak in the 5th layer and then start to decline.

Our results for Llama2-7B (Fig. 3a) show a similar trend, though less pronounced: the model correctly matches about 40% of corrupted words with the original, single-token, uncorrupted word.[8] We also note that the trend across layers persists, with accuracy peaking around layer 15.

## 4.2 MULTI-TOKEN WORDS

We have seen that LLMs are able to detokenize single-token words that are artificially broken into sub-words back into their correct (single-token) word. This indicates that these words are part of the model's inner lexicon. But what about multi-token words? By definition, these words do not have a single embedding vector in the BPE vocabulary, so we cannot compare them to any existing vector, and thus cannot use logit lens.

Instead, we use the Patchscopes technique (Ghandeharioun et al., 2024), which interprets the hidden representation of a model using the model's language abilities. In particular, we feed the model with the prompt "Repeat this word twice: 1) $X$ 2)", where $X$ is the hidden representation of the last word token. Our hypothesis is that if the multi-token word is found in the model's internal lexicon, then it will "understand" its representation in the input layer as well, and successfully repeat it by generating all tokens of the original word (see Fig. 1, right).

We evaluate the model by computing the rate of retrieval—the proportion of times it generates the correct (multi-token) word. We use the same models and dataset as in the previous experiments, though in this case we do not further split or add typos, as the words are already split into multiple tokens. Particularly, we feed each multi-token word (along with its 100-tokens context, as before) to the model, and extract the last token's representation from each layer. We then use Patchscopes' prompt in a new run to check if the model regenerates the original multi-token word.

Our Llama2-7B results (Fig. 3b) show a striking trend: when feeding the model with vectors from layers 5–7, it is able to repeat the word in 64% of the cases, despite never seeing this vector as input before.[9] As to the different layers, we observe a similar trend to the single-token results: retrieval rates increase rapidly at some point (here earlier than in previous experiments, around layer 3-4), reach a peak at layer 7, and then start to decline. Combined, these results suggest that the model treats multi-tokens words as if they were in the vocabulary but split into sub-word tokens, indicating a latent vocabulary that expands beyond the tokenizer's limitations. Interestingly, we observe that 22.6% of the multi-token words are never successfully decoded from any internal layer, hinting that they might not be represented in the model's inner lexicon.

---

[8]Here, 66% of the words are correctly mapped in at least one layer.
[9]Overall, 77.4% of the words are repeated correctly in at least one layer.

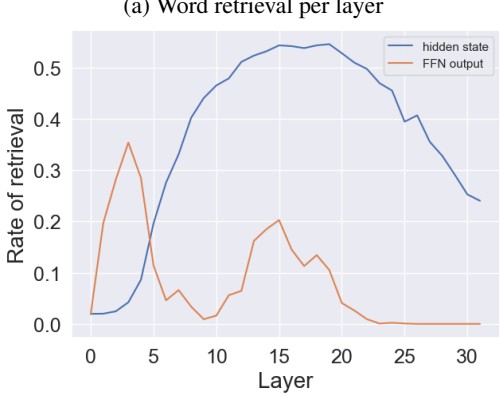 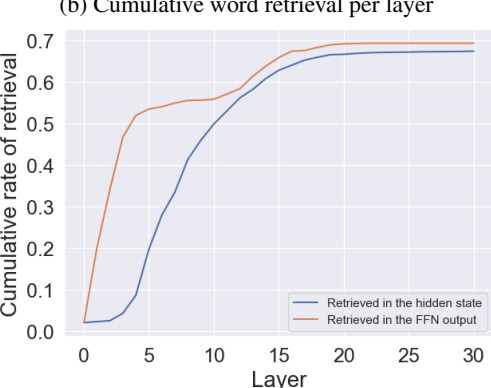

(a) Word retrieval per layer    (b) Cumulative word retrieval per layer

Figure 4: Word retrieval in FFN layers vs. in the hidden states. (4a) Retrieval rates across layers. The FFN values peak before word retrieval begins in the hidden layers. (4b) Its cumulative version, showing word retrieval occurs earlier and more frequently in FFNs than in the hidden representation.

## 5   HOW DOES DETOKENIZATION HAPPEN?

We have so far shown that LLMs perform a detokenization step: at some point, the hidden representation of the final token of a given word becomes strikingly similar to the (single-tokenx) vector of that word. This process is robust to artificial splits of single token words, to splits due to typos, and even to multi-token words, which the model can still recognize at the input layer, despite never having seen them there during training.

We next turn to ask **how** does the model reconstruct full word representations from sub-word tokens? We aim to understand the dynamics of this process by analyzing the main transformer components: feedforward network layers (FFN) and the attention mechanism.

### 5.1   WORD RETRIEVAL USING THE FEEDFORWARD MECHANISM

FFN layers have been shown to serve as key-value tables for storing memory (Geva et al., 2022; 2021; Meng et al., 2022b). We hypothesize that this memory might be used to store the inner lexicon as well. Particularly, we suggest that the model uses the FFNs to refine the representation of the final token, enabling it to retrieve the original word.

To test this hypothesis, we repeat the typos experiment of single-token words (Sec. 4.1), but this time applying logit lens to the *output of the FFN layers*, instead of the hidden representation. Figure 4a shows the retrieval rate of the original word. We observe that the retrieval of the full word concepts from the FFN layers occurs a few layers earlier compared to the hidden state, which indicates that they help refine it. Nonetheless, the rate of retrieval appears to be lower in FFNs, which suggests that other factors might come into play in building the hidden representation of words. However, in Fig. 4b we plot the cumulative retrieval rate, i.e., for each layer, whether the word is identified in *any* layer so far. Our results indicate a different conclusion: the FFN update vectors match the (single-token) input representation of the word 70% of the time in any layer, particularly both earlier and more frequently compared to the hidden representation. This hints that FFNs indeed play a substantial role in building the internal word representations in LLMs.

We further investigate the role FFNs play in detokenization through ablation experiments on FFN updates in Llama2-7B. Specifically, we test whether these updates are necessary for the word representation to emerge in the residual stream. To do so, we artificially split single-token words in a similar setting to Sec. 4.1, this time focusing on derived words, formed by adding a suffix to a root word.[10] We split these back to two parts, for example, we divide "eating" to "eat" and "ing", so that processing the suffix token is essential to reconstructing the word correctly. Using logit lens, we ex-

---

[10] We examine words with three common suffixes: "ing", "ion", and "est".

amine the FFN updates to the suffix token's residual stream, and selectively ablate those associated with the original single-token word (∼5% of layers). As a control, for each word, we ablate an equal number of random FFN updates. Our results (Fig. 12 in App. E) show that removing the updates carrying full-word representations dramatically reduces retrieval rates—from 85% without ablation to just 18%. In contrast, ablating random FFN updates has little to no effect.

This indicates that FFN updates are essential for a detokenized representation to emerge in the final token. But does this affect the model's ability to "understand" the word, particularly when predicting the next token? To investigate this, we evaluate the model's ability to retrieve the capital cities of countries, using the prompt "The capital of [COUNTRY] is _____". We use single-token country names artificially split into two tokens, and follow the previous ablation protocol. We observe notable patterns: randomly ablating FFN updates reduces performance from 88% to 74%,[11] likely due to disrupting the model's factual recall mechanism (Meng et al., 2022b). However, specifically canceling updates detected as country's name (similarly occurring in only ∼5% of layers) causes a sharp drop to 41%. Overall, our results suggest that FFNs are central to detokenization, and actively reconstruct full-word meanings in LLMs.

## 5.2 TOKEN AGGREGATION

Our results suggest that LLMs use the FFN layers to store and retrieve word representations, which are accessed in the final token to reconstruct the complete word. However, this role of the FFN layers only begins to emerge around layers 3–4. **What happens earlier**? We build on previous work showing that early layers primarily integrate information from nearby tokens to compose entities (Lad et al., 2024), and hypothesize that the model starts by aggregating information from the previous sub-word tokens.

To test this, we extract all two-token words in a subset of 1,000 WIKITEXT-103 documents (a total of 5,571 words), and feed them, along with their context, to Llama2-7B. We then measure the average attention weights of the final token to the prefix token in each layer. As a control, we also measure the average attention weights assigned by single-token words to their preceding token.

Our results (Fig. 5) support previous findings (Lad et al., 2024)—the attention to previous tokens is high in the first 2–3 layers, but then declines sharply (by up to ∼90%).[12] For single-token words, we observe a similar attention pattern to the previous token in the first layers, but importantly—the initial peak is significantly lower than in multi-token words.[13] Still, in later layers, the attention weights of single-token words are in turn *higher* than for multi-token words. These results suggest LLMs strongly attend to the preceding sub-word tokens of multi-token words *at first*, but then largely ignore them.

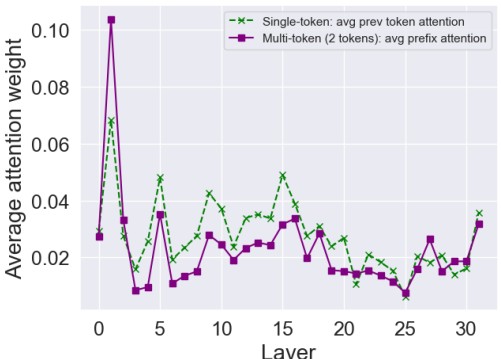

Figure 5: Attention weights for 2-token words: Early peaks (layers 2–3) show high attention values from the second sub-word token to the first, but these values decline rapidly. Attention from single-token words to their previous token shows a similar trend, though with substantially lower values at first, which become higher later.

Altogether, our results suggest that LLMs perform a detokenization process by first aggregating information from the prefix tokens into the final token's hidden representation, and then refining the representation of the final token using the FFN layers to retrieve the full word's concept representation. This two-stage process of token aggregation and concept retrieval provides insight into the mechanisms LLMs use to handle sub-word tokens and reconstruct word-level representations.

---

[11]We note that baseline performance when passing country names without artificial splits is 95%.

[12]Experiments with 3- and 4-token words show a similar trend, see App. B.

[13]The diverging patterns between single and multi-token words are statistically significant; see App. B.

## 6    EXPANDING LLM VOCABULARY WITHOUT FINETUNING

We have shown that language models internally fuse multi-token words into a single-token representation, and that they can further "read" these representations as inputs, and decode the original multi-token words. This raises the question: can models use these fused representations instead of the original multi-token inputs—to encode input prompts using less tokens and reduce computation? Similarly, models were shown to implicitly predict several future tokens in a single hidden state without being explicitly trained to do so (Pal et al., 2023); can we leverage these representations to enable models to predict multi-token words in a single inference step?

Motivated by our findings, we explore whether we can expand the model's vocabulary with new input and output embeddings for originally multi-token words, without any updates to model parameters.[14] This goal is of practical importance, especially for low-resource languages and domains: even tokenizers built for multilingual support often produce substantially longer token sequences for non-English languages—up to 13 times longer than equivalent English texts—impacting inference cost and speed (Ahia et al., 2023; Sengupta et al., 2023; Petrov et al., 2023). Still, prior attempts to expand tokenizer vocabulary post-hoc are limited in number, and require substantial additional training (Kim et al., 2024; Zhao et al., 2024). In contrast, we propose to detect words the model "knows" and successfully detokenizes to a single vector, and use this to obtain new token embeddings.

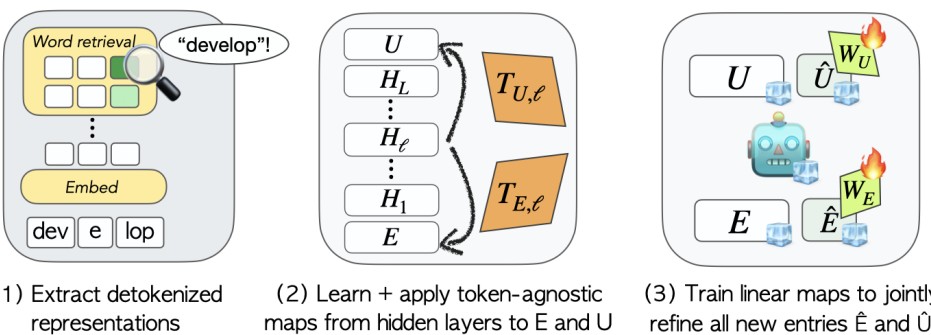

(1) Extract detokenized representations

(2) Learn + apply token-agnostic maps from hidden layers to E and U

(3) Train linear maps to jointly refine all new entries Ê and Û

Figure 6: Our 3-step method to expand LLM vocabulary without updates to core model parameters.

Our framework for vocabulary expansion follows a 3-step process (Fig. 6). Given a multi-token word $w$ that we would like to add to the vocabulary, and the model's original input embedding and output unembedding matrices $E$ and $U$ with hidden dimension $d$, we (1) extract a detokenized, single-token representation $r$ for $w$ using a similar approach to Sec. 4.2: pass $w_i$ as input to the model and apply PATCHSCOPES (Ghandeharioun et al., 2024) with prompt $P$[15] to the last token's hidden states at all layers. We then identify $\ell$, the earliest layer at which the hidden state is successfully decoded into the full word, and set $r$ as that hidden state. Next, we (2.1) learn a set of linear maps $T_{\ell,E}$ and $T_{\ell,U}$ to project hidden states from layer $\ell$ of the model to the embedding and unembedding spaces, based only on the *existing* in-vocabulary tokens.[16] Then, (2.2) for each detokenized representation $r$ taken from layer $\ell$, we apply $T_{\ell,E}$ and $T_{\ell,U}$ to obtain the initial entries in the embedding and unembedding matrices to represent the word, $\hat{e}$ and $\hat{u}$. Finally, after computing the initial entries for all new words, we (3) refine the new representations to obtain the final entries: we initialize two $d \times d$ all-zeros matrices $W_E$ and $W_U$, and set the new entries as $e = \hat{e} + W_E \hat{e}$ and $u = \hat{u} + W_U \hat{u}$. We train the refinement matrices $W_E$ and $W_U$ jointly in a short continued pretraining run, while keeping all other parameters frozen. Finally, we compute the final $e$ and $u$, and use these to represent the new word in the expanded vocabulary. Importantly, if a word is never successfully decoded from the hidden states in any of the layers in step (1), we assume it is not found in the model's inner lexicon, and therefore do not add it to the vocabulary.

---

[14]We do add new entries to the model's embedding and unembedding matrices, but these are intrinsically required to expand the vocabulary and represent the new vocabulary words.

[15]For $P$, we use the template "x x x x" where x is the patched representation of the new word.

[16]We learn $T_{\ell,E}$ and $T_{\ell,U}$ by fitting an orthogonal procrustes transformation from the layer $\ell$ hidden states of all in-vocabulary tokens (when passed as inputs on their own), to their corresponding embeddings or unembeddings (see App. G for details). Importantly, these projections only rely on the model's existing vocabulary, and do not depend on which multi-token words we choose to add to the vocabulary.

|  |  | WIKITEXT-103 | | | PUBMED | | | WIKI40B-Arabic | | |
|---|---|---|---|---|---|---|---|---|---|---|
|  |  | Original | Mean Embed. | Ours | Original | Mean Embed. | Ours | Original | Mean Embed. | Ours |
| New words | *new* token | – | 0.071 | 0.171 | – | 0.123 | 0.180 | – | 0.117 | 0.402 |
|  | *original* or *new* token | 0.322 | 0.178 | 0.284 | 0.280 | 0.171 | 0.259 | 0.413 | 0.119 | 0.414 |
| All words | | 0.522 | 0.473 | 0.519 | 0.517 | 0.479 | 0.511 | 0.535 | 0.211 | 0.532 |

Table 1: Token-level accuracy of finetuning-free vocabulary expansion for Llama2-7B on three datasets. We compare our method using detokenized representations (*Ours*) to the original model (*Original*) and to an expansion baseline using the word's average token embeddings from $E$ and $U$ (*Mean Embed.*; Gee et al. 2022). Accuracy is reported for newly added words, where we distinguish between correctly predicting the new tokens (top row) and predicting either the word's original first token, as the unexpanded model would, or its new token (middle row). We also report overall performance on all tokens (last row). Our method enables the frozen model to use the newly added input and output embeddings, while maintaining overall model performance.

We apply our approach to LLAMA2-7B and experiment with three datasets: WIKITEXT-103 (Merity et al., 2017), abstracts of biomedical articles from PUBMED (Xiong et al., 2024), and the Arabic split of WIKI40B (Guo et al., 2020). For each dataset, we expand the model's vocabulary with all multi-token words that appear at least $m$ times in the test set.[17] We then learn the refinement matrices $W_E$ and $W_U$ using 20M tokens from the train set.[18] We evaluate models in a next-word prediction setup and compare our approach against two baselines: the original model with an unmodified vocabulary (*original*); and an expanded model that follows our framework but uses the mean embeddings of the word's tokens in $E$ and $U$ to initialize its new representation, instead of the detokenized representations $r$, following Gee et al. (2022; *mean embedding*). For evaluation, perplexity is not a suitable metric, as differences in vocabulary between the approaches skew perplexity scores and prevent a fair comparison. Instead, we measure the model's token-level top-1 accuracy when it is given each token's previous context.

Our results (Table 1) indicate that models can generalize to new vocabulary entries surprisingly well when these are initialized with their own detokenized representations. Unlike the *mean embeddings* expansion baseline, which both struggles with using new tokens and degrades overall performance, our method allows the model to successfully integrate new vocabulary words while preserving its accuracy on existing words. This effect is particularly pronounced in Arabic WIKI40B, where the model almost always selects new tokens instead of the original tokenization (40.2% vs 41.3%), demonstrating the potential of our approach for multilingual and domain-specific applications.

Our method also shows potential to reduce the number of tokens processed during encoding and inference: across the three datasets, we observe a reduction in the total number of tokens processed during encoding by 10.5% to 14.5% (see Table 5 in App. H), while maintaining model performance. In future work, we will further explore the application of our framework to multilingual adaptation (Petrov et al., 2023; Alabi et al., 2022) and continual domain-adaptive pretraining (Ke et al., 2023; Yıldız et al., 2024; Gururangan et al., 2020).

## 7  CONCLUSION

The ability of LLMs to comprehend and generate language relies on intricate internal processes, and understanding these mechanisms is crucial for improving model performance and efficiency. In this work, we unraveled the word detokenization process, shedding light on how models internally transform fragmented sub-word tokens into coherent word representations formed at the last token. Our results indicate that this mechanism manifests in early to middle layers, where models attempt to reconstruct words by mapping them to an inner lexicon using their FFN layers. We provided evidence this lexicon is more exhaustive than the tokenizer's vocabulary, and could help models to recognize words even amidst noise.

Our work also unlocks practical avenues for optimizing tokenization, as well as the speed and cost of inference. We demonstrated one such application and presented a finetuning-free method to expand the vocabulary of LLMs. We hope our work paves the way for more efficient and versatile models.

---

[17]We use $m = 1, 5, 50$ for WIKITEXT-103, PUBMED and Arabic WIKI40B respectively.

[18]We use a sequence length of 512 and train on 10,000 sequences, taking up to 30 minutes on a single GPU.

ACKNOWLEDGMENTS

This work was supported in part by the Israel Science Foundation (grant no. 2045/21) and by NSF-BSF grant 2020793. We are grateful to Iddo Yosha for his valuable insights and thank the anonymous reviewers for their constructive feedback.

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

## A  WORDS VS. NONWORDS ANALYSIS

To further analyze our results from Sec. 3, we conduct a failure analysis, focusing on false negatives (FN) and false positives (FP). Our analysis indicates that the former—valid words misclassified as gibberish—are often rare and complex words, suggesting that the model's internal vocabulary may lack representations for infrequent words. On the other hand, false positives—where gibberish is misclassified as a word—typically involves nonwords that closely resemble valid words, likely due to shared sub-word structures. See Tab. 2 for a few examples.

| Original Word | Predicted Value | True Value | Status |
|---|---|---|---|
| _Unitarianism | gibberish | word | FN |
| _killy | gibberish | word | FN |
| _quadruropic | word | gibberish | FP |
| _nonwith | word | gibberish | FP |

Table 2: Examples of false negative (FN) and false positive (FP) for the word vs. nonword experiment (Sec. 3).

We further conduct additional experiments to compare the performance of our custom nonword dataset with the linguistically-motivated ARC Nonword Database (Rastle et al., 2002), which contains morphologically plausible nonwords designed to resemble real words to humans. Our results (Fig. 7) indicate a higher differentiability on the ARC dataset compared to our own dataset, which highlights that our nonword creation procedure effectively mitigates potential biases. This result strengthens our conclusions: the last token provides significant cues for distinguishing words from nonwords, particularly in the middle layers of the model after several layers of processing.

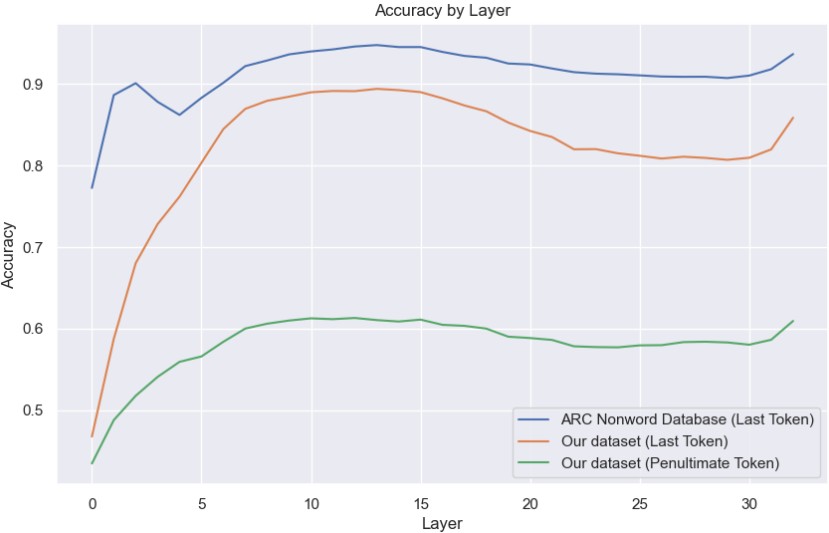

Figure 7: Accuracy comparison by layer for distinguishing words from nonwords. The ARC Non-word Database (blue line) exhibits the highest accuracy across layers, potentially due to its morphologically plausible nonwords. In contrast, our dataset (orange line, last token) achieves slightly lower accuracy, demonstrating the effectiveness of our bias-mitigated nonword generation process. The penultimate token from our dataset (green line) shows significantly lower accuracy, highlighting the importance of the last token in distinguishing words from nonwords.

In another experiment, we evaluate the penultimate token representation against nonword tokens for words of length three or more tokens. For example, in a word like "unhappiness," we use the inner token "h" (e.g., *un-h-appiness*). Unlike the final tokens, penultimate tokens are poorly distinguishable from nonwords, achieving significantly lower accuracies. These results suggest that the model's

ability to separate words from nonwords is highly dependent on the position and completeness of token representations.

Finally, we expand the comparison of our dataset across three additional models—Llama3-8B, Mistral-7B, and Yi-6B. Our results across these models (Fig. 8) mirror the trends observed with Llama2-7B, with accuracy improving as layers deepened but peaking and gradually declining in later layers. This consistency further supports our findings that the model's ability to classify words vs. nonwords relies on nuanced patterns in internal token representations, unaffected by co-occurrence biases or morphological artifacts.

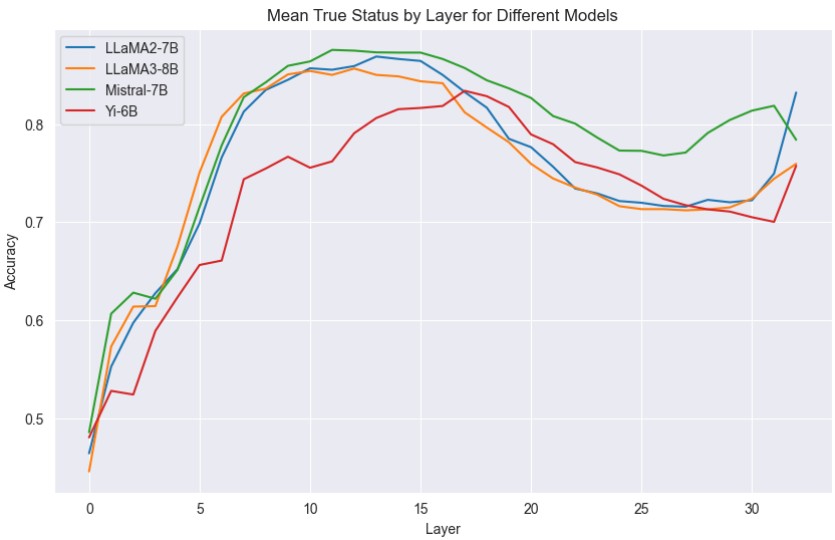

Figure 8: Mean true status accuracy by layer for distinguishing words from nonwords across multiple models (Llama2-7B, Llama3-8B, Mistral-7B, Yi-6B) using our dataset. All models exhibit similar trends: accuracy improves across initial layers, peaks in the middle layers, and declines in deeper layers. This pattern demonstrates the robustness of our findings across different model architectures and sizes.

## B    STATISTICAL ANALYSIS OF TOKEN AGGREGATION IN MULTI-TOKEN WORDS

We repeat the multi-token experiments Sec. 5.2 for 3- and 4-token words. Our results (Fig. 9) show a very similar trend to 2-token words (Fig. 5).

We further present the detailed statistical analysis conducted of our experiments in Sec. 5.2 to examine **token aggregation** in multi-token words compared to single-token words (control group). The objective is to investigate whether the model exhibits significantly different attention patterns between these groups, indicative of the detokenization process.

To determine significance, we perform two one-sided t-tests per layer for the Llama2-7B case: (1) Testing whether attention to prefix tokens in multi-token words is higher than to previous tokens in single-token words. (2) Testing whether attention to previous tokens in single-token words is higher than to prefix tokens in multi-token words. Table 3 shows the p-values and significance levels for each layer. Significance levels are denoted as ns (not significant), and *** ($p < 0.001$).

The results reveal that in layers 1 and 2, attention to prefix tokens in multi-token words is significantly higher than in single-token words, suggesting the early phase of **Token Aggregation**. From Layers 3 to 17, attention to single-token words is higher, indicating a shift in attention focus from prefix attention in relation regular close tokens. Notably, there are intermittent increases in attention to prefix tokens at Layers 18, 21, 25, 27, 29, and 30, possibly signaling a transfer into a prediction ensembling phase in which the attention in general is less important (Ben Artzy & Schwartz, 2024) and therefore we don't see a coherent pattern.

| Layer | p-value (Multi > Single) | p-value (Single > Multi) | Significance |
|---|---|---|---|
| 0 | $9.999 \times 10^{-1}$ | $5.891 \times 10^{-5}$ | Single-token > Multi-token (***) |
| 1 | 0.000 | 1.000 | Multi-token > Single-token (***) |
| 2 | 0.000 | 1.000 | Multi-token > Single-token (***) |
| 3 | 1.000 | $1.252 \times 10^{-273}$ | Single-token > Multi-token (***) |
| 4 | 1.000 | 0.000 | Single-token > Multi-token (***) |
| 5 | 1.000 | $4.466 \times 10^{-242}$ | Single-token > Multi-token (***) |
| 6 | 1.000 | 0.000 | Single-token > Multi-token (***) |
| 7 | 1.000 | 0.000 | Single-token > Multi-token (***) |
| 8 | 1.000 | 0.000 | Single-token > Multi-token (***) |
| 9 | 1.000 | 0.000 | Single-token > Multi-token (***) |
| 10 | 1.000 | 0.000 | Single-token > Multi-token (***) |
| 11 | 1.000 | $7.847 \times 10^{-87}$ | Single-token > Multi-token (***) |
| 12 | 1.000 | $6.307 \times 10^{-303}$ | Single-token > Multi-token (***) |
| 13 | 1.000 | $1.417 \times 10^{-255}$ | Single-token > Multi-token (***) |
| 14 | 1.000 | $2.397 \times 10^{-172}$ | Single-token > Multi-token (***) |
| 15 | 1.000 | 0.000 | Single-token > Multi-token (***) |
| 16 | 1.000 | $5.857 \times 10^{-25}$ | Single-token > Multi-token (***) |
| 17 | 1.000 | $1.414 \times 10^{-152}$ | Single-token > Multi-token (***) |
| 18 | $5.646 \times 10^{-5}$ | $9.999 \times 10^{-1}$ | Multi-token > Single-token (***) |
| 19 | 1.000 | $7.919 \times 10^{-161}$ | Single-token > Multi-token (***) |
| 20 | 1.000 | $2.036 \times 10^{-286}$ | Single-token > Multi-token (***) |
| 21 | $1.399 \times 10^{-190}$ | 1.000 | Multi-token > Single-token (***) |
| 22 | 1.000 | $3.647 \times 10^{-44}$ | Single-token > Multi-token (***) |
| 23 | 1.000 | $6.008 \times 10^{-13}$ | Single-token > Multi-token (***) |
| 24 | 1.000 | $1.062 \times 10^{-15}$ | Single-token > Multi-token (***) |
| 25 | $5.274 \times 10^{-68}$ | 1.000 | Multi-token > Single-token (***) |
| 26 | 1.000 | $3.670 \times 10^{-12}$ | Single-token > Multi-token (***) |
| 27 | 0.000 | 1.000 | Multi-token > Single-token (***) |
| 28 | 1.000 | $2.446 \times 10^{-10}$ | Single-token > Multi-token (***) |
| 29 | $1.438 \times 10^{-202}$ | 1.000 | Multi-token > Single-token (***) |
| 30 | $9.506 \times 10^{-99}$ | 1.000 | Multi-token > Single-token (***) |
| 31 | $6.731 \times 10^{-1}$ | $3.270 \times 10^{-1}$ | Not significant (ns) |

Table 3: Results of one-sided t-tests comparing attention weights between multi-token and single-token words across layers over Llama2-7B model.

(a) Llama2-7B Attention weights for 3-tokens words    (b) Llama2-7B Attention weights for 4-tokens words

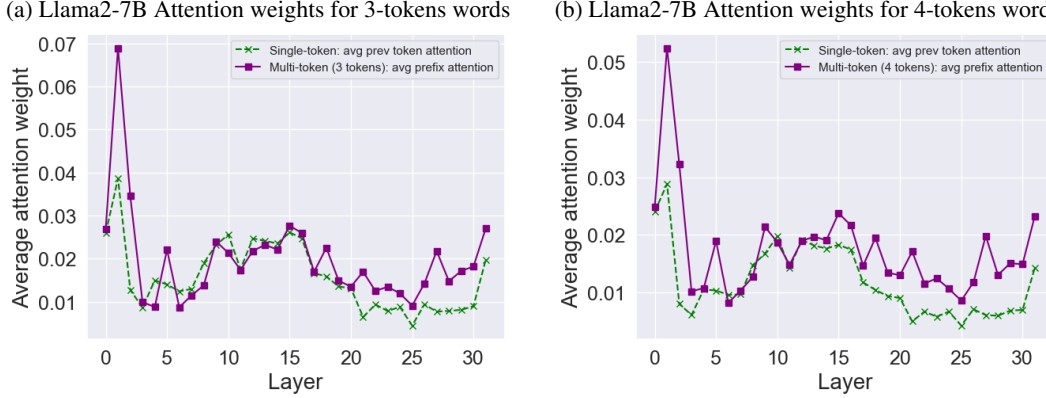

Figure 9: Analysis for 3- and 4-token words for Llama2-7B. The higher attention pattern in layers 1-2, while lower values are observed afterwards, is consistent with our results for 2-token words (Fig. 5).

In conclusion, the attention mechanism differs significantly between multi-token and single-token words, indicating that the detokenization process involves initial amplification of attention to sub-word tokens followed by a reduction as the model obtains whole-word representations.

## C    WORD RETRIEVAL FOR SINGLE-TOKEN AND MULTI-TOKEN WORDS ACROSS MODELS

This section presents a detailed comparison of word retrieval performance (Sec. 4) across several models (Llama2-7B, Llama3-8B, Yi-6B, and Mistral-7B) for both single-token and multi-token words. The evaluation focuses on how effectively each model retrieves the original word across layers, especially in challenging cases like artificially separated single-token words, typos, and multi-token words.

Across all experiments (Fig. 10), we observe a similar trend where the retrieval rate increases over the first several layers, peaks around the middle layers, and then decreases in the later layers. The main difference across models lies in the peak performance, especially in cases involving typos and multi-token words, where more advanced models such as Llama3-8B and Mistral-7B demonstrate superior performance in reconstructing the original word representations.

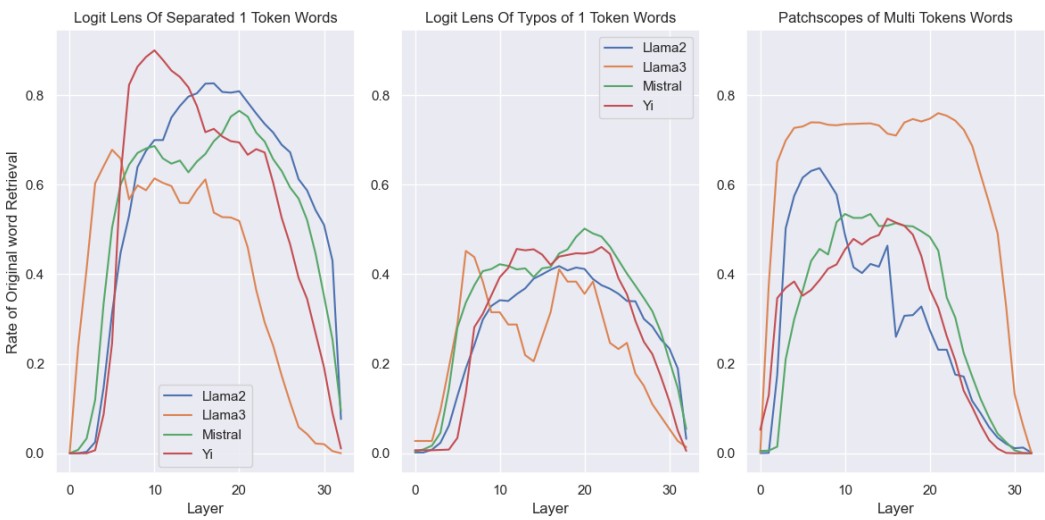

Figure 10: Layer-wise word retrieval rates for single-token and multi-token words across all models.

### C.1    CUMULATIVE WORD RETRIEVAL FOR SINGLE-TOKEN AND MULTI-TOKEN WORDS

Fig. 11 shows the results of cumulative word retrieval (Sec. 4) across various models, focusing on both single-token and multi-token words. For each model, we analyze the ability of the LLM to retrieve the original word from sub-word tokens across its layers. The cumulative retrieval is calculated as the proportion of words that are successfully retrieved at each layer, with the percentage increasing as more words are recovered throughout the model's layers.

**Single-token words**    The cumulative word retrieval for single-token words—those that are artificially split into sub-word tokens (via typographical errors or manual splits)—shows a rapid increase in retrieval success in the early layers. For Llama2-7B, for instance, cumulative retrieval reaches 93.2% for words split by manual intervention, and 66% for words affected by typos by the middle layers. This pattern is observed across models, with retrieval generally peaking around layers 15-20.

**Multi-token words**    For multi-token words, which are naturally split due to being out-of-vocabulary for the tokenizer, the cumulative retrieval process follows a similar trajectory. However, in models like Llama2-7B, the retrieval peaks earlier in the model, with a cumulative retrieval rate of 77.41%. Other models like Llama3 and Yi show higher cumulative retrieval rates, suggesting improved efficiency in handling multi-token words, potentially due to larger model capacities and internal dictionaries.

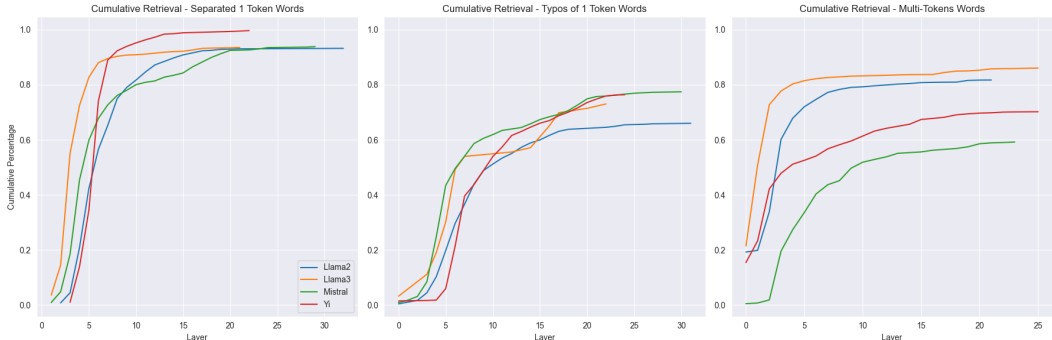

Figure 11: Cumulative word retrieval for single-token and multi-token words across all models.

The similarity in cumulative retrieval between single-token and multi-token words suggests that LLMs treat out-of-vocabulary words in a manner similar to sub-word-tokenized words, accessing a latent vocabulary to reconstruct full word representations.

## D    INTRODUCING TYPOS FOR SINGLE-TOKEN WORDS

In this section, we describe the process of introducing typos into single-token words to split them into multiple tokens (Sec. 4.1). The modification applies to words longer than four characters and involves randomly performing one of three operations: substituting two characters, deleting a character, or inserting a new character. By introducing these slight variations, the word becomes unfamiliar to the tokenizer, causing it to be divided into multiple smaller tokens during tokenization. Particularly, this process results in splitting words into 2–5 tokens. Table 4 shows examples of the different splits.

| Description | perturbed | new tokens |
|---|---|---|
| Substitution of two characters | devel**po**ment | ['de', 'vel', 'p', 'oment'] |
| Deletion of one character | develment | ['de', 'vel', 'oment'] |
| Insertion of one character | devel**f**opment | ['dev', 'elf', 'op', 'ment'] |

Table 4:  Examples of the different typos we consider, exemplified by perturbing the single-token word "development" (Sec. 4.1).

## E    ABLATION EXPERIMENT ON SUFFIX-SPLIT WORDS

In Sec. 5, to test the role of FFN layers in detokenization, we conduct an intervention-based experiment measuring how ablations to FFN updates affect word retrieval. We run this experiment on all single-token words from WIKITEXT-103 that end with one of three common suffixes: *"ing," "ion,"* or *"est"*. Each word is then artificially split to two parts—the root word and the suffix. For example, the word *"eating"* is split into *"eat"* and *"ing"*, while *"connection"* is split into *"connect"* and *"ion"*. This ensures that processing the suffix token is necessary to reconstruct the identity of the word, and reduces possible effects of strong distributional artifacts of token co-occurrence.

Using logit lens, we identify FFN layers where the update to the residual stream can decoded as the original single-token word. We then ablate these layers by zeroing out their updates to the residual stream. As a control, we ablate the same proportion (5%) of random FFN layers.

Our results, shown in Fig. 12, reveal that ablating the identified FFN layers lead to a sharp drop in word retrieval rates, effectively disrupting the detokenization process. In contrast, ablating random layers has little to no effect on retrieval accuracy. This indicates that the FFN layers play a critical role in reconstructing word representations rather than simply enhancing contextualization.

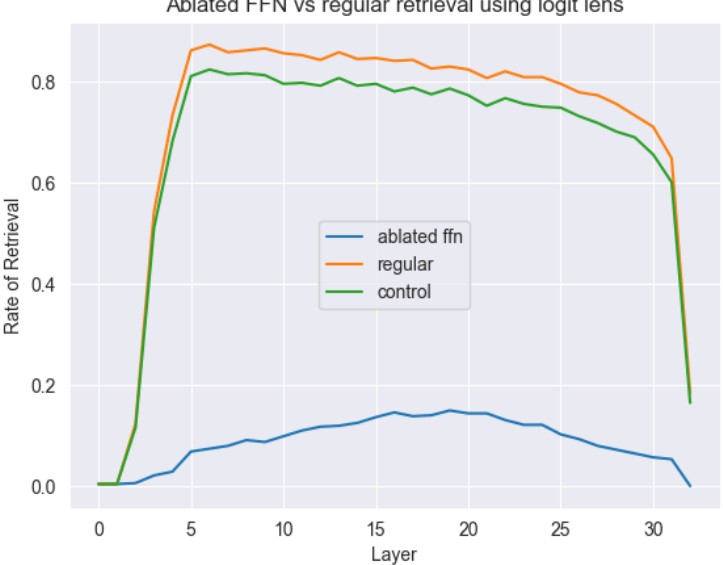

Figure 12: Comparison of retrieval rates using logit lens for suffix-split words under three conditions: ablation of identified FFN layers (blue line), regular retrieval (orange line), and control with random FFN layer ablation (green line). Ablation of critical layers causes a sharp drop in retrieval accuracy, highlighting their importance in detokenization.

## F  COMPARISION BETWEEN LOGIT LENS AND COSINE SIMILARITY

To validate our use of the logit lens, we repeat the artificial split-word experiment using cosine similarity. We use the same setting of artifical splits based on suffixes detailed above in App. E.

For the logit lens, we adapt the standard implementation to measure similarity between the hidden representation of the final token and the input embedding space (vocabulary space), rather than the output embedding space typically used to predict the next token. This adjustment allows us to directly evaluate the alignment between the hidden states and the embeddings of the original words in the vocabulary. In parallel, cosine similarity is used to compute the similarity between the same hidden states and the embeddings of the original words in the vocabulary space.

Our results, shown in Fig. 13, demonstrate that both methods produce nearly identical patterns. In both cases, retrieval accuracy peaks in the middle layers, where the hidden representation of the suffix token aligns most closely with the original word. Beyond these middle layers, retrieval accuracy declines, likely reflecting the model's shift toward representing the prediction of the next token rather than maintaining the full representation of the current word.

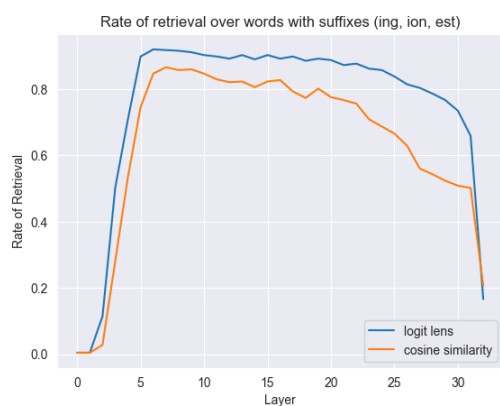

Figure 13: Comparison of retrieval accuracy for split-word experiments using either logit lens (blue line) and cosine similarity (orange line) across model layers.

These findings reinforce the validity of using the logit lens when adapted to the input embedding space, as a streamlined approach for analyzing model representations. This approach results in similar trends to cosine similarity while offering a more interpretable and direct framework for measuring alignment with the vocabulary space.

# G   LEARNING THE LINEAR MAPS $T_{\ell,E}$ AND $T_{\ell,U}$

To expand the model's vocabulary without modifying its core parameters, we construct linear transformations that map hidden states at different layers to the model's embedding and unembedding spaces. By learning these transformations solely from the model's existing vocabulary, we can infer new token representations without modifying any of the model's core weights. This section details our method for learning these mappings.

## G.1   EXTRACTING MODEL REPRESENTATIONS

Given a pretrained language model with input embedding matrix $E \in \mathbb{R}^{V \times d}$ and output unembedding (LM head) matrix $U \in \mathbb{R}^{V \times d}$, where $V$ is the vocabulary size and $d$ is the hidden dimension, we extract the representations used for learning the transformations as follows:

- **Embedding and Unembedding Matrices**: We extract the model's embedding matrix $E$ and LM head matrix $U$ before making any modifications to the vocabulary.
- **Hidden States Across Layers**: For each token $t$ in the model's original vocabulary, we pass $t$ as a single-token input and record its hidden state at every layer of the model. Let $h_\ell(t)$ denote the hidden state of token $t$ at layer $\ell$.

## G.2   LEARNING THE LINEAR MAPPINGS

For each layer $\ell$, we aim to learn two linear transformations:

- $T_{\ell,E}$ that maps hidden states to input embeddings.
- $T_{\ell,U}$ that maps hidden states to unembedding representations.

We learn these mappings as **orthogonal Procrustes problems** (Schönemann, 1966), which seek to find the best orthogonal transformation aligning two sets of vectors. Specifically, for each layer $\ell$, we solve:

$$T_{\ell,E} = \arg\min_T \sum_{t \in V} \|Th_\ell(t) - e_t\|^2, \quad \text{subject to } T^\top T = I \tag{1}$$

$$T_{\ell,U} = \arg\min_T \sum_{t \in V} \|Th_\ell(t) - u_t\|^2, \quad \text{subject to } T^\top T = I \tag{2}$$

where $e_t$ and $u_t$ are the embedding and unembedding vectors of token $t$, respectively. The constraints enforce that each transformation preserves distances and does not distort the structure of the space. In our experiments, we use the Python implementation of Meng et al. (2022a).

## G.3   NORMALIZATION WITH RMS SCALING

To preserve the relative magnitudes of embedding and unembedding entries, we normalize all representations using their **root mean square (RMS) norm** (Zhang & Sennrich, 2019). Specifically:

1. **Preprocessing Training Representations**: Before fitting the Procrustes transformations, we normalize all hidden states, embeddings, and unembedding vectors by dividing each vector $x$ by its RMS norm:

$$x_{\text{norm}} = \frac{x}{\|x\|_{\text{RMS}}}, \quad \text{where } \|x\|_{\text{RMS}} = \sqrt{\frac{1}{d} \sum_{i=1}^{d} x_i^2}. \tag{3}$$

2. **Applying the Learned Maps**: When using the trained transformations on new detokenized representations $r$, we apply the following steps:
   (a) Normalize $r$ by its RMS norm: $r_{\text{norm}} = \frac{r}{\|r\|_{\text{RMS}}}$.
   (b) Apply the learned transformation: $\hat{e} = T_{\ell,E} r_{\text{norm}}$, $\hat{u} = T_{\ell,U} r_{\text{norm}}$.
   (c) Rescale by the mean RMS of the target space:

$$e = \hat{e} \cdot \mathbb{E}[\|e_t\|_{\text{RMS}}], \quad u = \hat{u} \cdot \mathbb{E}[\|u_t\|_{\text{RMS}}]. \tag{4}$$

This ensures that the new embeddings and unembeddings maintain a scale consistent with the original vocabulary.

## H   EFFICIENCY GAINS FROM VOCABULARY EXPANSION

Beyond improving model performance on newly added words, our vocabulary expansion method directly reduces the number of tokens required to encode input text, and could lead to further potential efficiency gains in inference.

Table 5 summarizes the average reductions in sequence length achieved by encoding texts from each domain using the expanded vocabulary instead of the original vocabulary in the three datasets. Using our method, we attempt to expand Llama2-7B's vocabulary with all multi-token words appearing at least $m$ times in the test set, where we use $m = 1, 5, 50$ for WIKITEXT-103, PUBMED, and Arabic WIKI40B, respectively.

| Dataset | # Attempted | # New Words | Token Reduction |
|---|---|---|---|
| WIKITEXT-103 | 14.1k | 10.1k | 10.5% |
| PUBMED | 9.5k | 5.4k | 13.5% |
| WIKI40B-Arabic | 4.4k | 0.7k | 14.5% |

Table 5:   Reduction in average sequence length when encoding text with the expanded vocabulary for Llama2-7B. *# Attempted* is the number of multi-token words tested for expansion, while *# New Words* is those detected as detokenized using Patchscopes.

We find that the token reduction rates depend on the success rate of extracting detokenized representations using Patchscopes (which in turn depend on the textual domain and language). WIKITEXT-103 achieves a 10.5% reduction in token count, with 72.9% of attempted words successfully converted into new vocabulary entries. PUBMED shows a higher token savings rate (13.5%) despite a lower success rate (58.3%), as its vocabulary expansion targets more domain-specific multi-token terms. The largest efficiency gain is observed in Arabic WIKI40B, where encoding with the expanded vocabulary reduces token count by 14.5%, highlighting the method's potential for languages with inherently longer token sequences. However, the success rate of identifying detokenized representations in WIKI40B is lower (16.0%), suggesting room for improvement in expanding vocabularies for morphologically rich languages, and suggesting many Arabic words are not stored in the model's inner lexicon.

These results suggest that post-hoc vocabulary expansion can significantly reduce the computational cost of inference, particularly for non-English languages and domain-specific texts, without requiring any modifications to the model's core parameters.

