# OpenReview forum: "From Tokens to Words: On the Inner Lexicon of LLMs"
_ICLR.cc/2025/Conference — ICLR 2025 Poster_

### Official Review · Reviewer_t1G5 · 2024-10-29

**Soundness:** 2
**Presentation:** 3
**Contribution:** 3
**Rating:** 6
**Confidence:** 3

**Summary:**

The paper studies how subwords are detokenized through transformers into the original word for further processing and understanding (where LLM is able to distinguish between words and non-words are shown are shown in a preliminary study in this work). In this research direction, the paper makes the following contributions:
- The paper shows that detoenization process happens in the beginning-middle layers using techniques using techniques such as logic lens (single token) and patchscope (multi-token)
- The paper then carry on experiments suggesting that the detokenization happens within FFN layers
- Leveraging the above results, the paper shows that that transformer efficiency can be enhanced by introducing "decodable" token embeddings; the paper examines both input embeddings and output embeddings.

**Strengths:**

The paper presents a significant amount of content and materials while being easy to follow. The paper incorporates appropriately the related works so that it is relatively straightforward to situate the paper in the literature. Concretely, I think the paper has made the following contributions:
- Through techniques such as logic lens and patchscope, the paper demonstrates convincingly where the model performs the detokenization process by presenting clearly such studies.
- The paper shows that FFN serves to combine subword information in section 5.
- In the final section, the paper shows how the understanding can help transformer decoding in practice. The paper adds word embeddings both in input matrix and output matrix and show that the model can accelerate the inference while maintaining a good performance.

**Weaknesses:**

While the paper presents a complete study (with no missing component) in the detokenization study, it feels that the paper can still be further enhanced with some more in-depth studies, some of the questions I have put in the questions section but in general:
- The cumulative curve shows that FFN indeed contributes to detokenization. What about other components? Are there any hints/patterns that the authors observe in the detokenization process (e.g. what words are first detokenized)?
- Cumulative rate saturates at around 0.7 shown in the figure. What about the rest 30%? Are these limitations for the measured model? Do better models (e.g. llama3) perform better at these?
- More details will help section 6 and I list some of them in the questions section. I think these are just some of the questions that a common reader would have after reading the paper. I think the results in this section may be of practical importance and deserve to be enhanced with more empirical results.

**Questions:**

For section 5, the feedforward mechanism, why only FFN output are measured please? Does it make sense to also measure the residual part please?
For section 6:
- For all original vocabulary tokens, all models perform less well than the model with original vocabulary? Are there examples to illustrate these and some hints where models fall short?
- What would be the full newly token accuracy for the model with original vocabulary?
- With such techniques, what would be an estimated inference speed gain? For input embedding as well as for output embedding?

---

> ### Author Response · Authors · 2024-11-19
>
> We thank the reviewer for their detailed comments. We appreciate their acknowledgement of our main contributions, and for their appraisal of our paper as being clearly written.
>
>
> **Other components in the detokenization process**
>
> In terms of model components, as we show in Section 4.1, the attention mechanism plays a critical role in the detokenization process alongside the FFN. As noted by ChVq, prior work has also identified specific attention heads responsible for attending to sub-word tokens within BPE-tokenized words, further highlighting the importance of attention in this process.
>
> Beyond the mechanistic properties of the model, data-related factors also influence the rate of retrieval. For example, our experiments show that retrieval rates are higher for common words from widely used corpora like Wikipedia compared to less common words from older texts in PG19. This suggests that the frequency and distribution of words in the training data significantly impact their retrieval success.
>
> **Cumulative rate saturates at around 0.7**
>
> The saturation of the cumulative rate at around 0.7 aligns with the results in Section 4.1 and is largely due to noise introduced by typos. These typos cause about 30% of the words to remain unidentified despite the model having these words in its inner lexicon.
>
> Regarding the comparison with other models, we conducted additional experiments to compare pure retrieval across different models (see Fig. 9 in appendix D) . The cumulative subplots indeed show that Llama3 performs better than Llama2 in multi-token retrieval. However, for typos and artificial separations, both models exhibit similar results.
>
>
> **Question on section 5: FFN vs. residual stream**
>
> Both the FFN (orange line) and the residual stream (blue line) are measured and shown in Fig. 4.b.
>
> **Questions on section 6**
>
> Thank you for these great questions. We are actively working on addressing them and will provide updates once we have them:
>
> 1. **Performance on the original vocabulary tokens:** We are conducting an error analysis to identify where models fall short compared to those with the original vocabulary. This includes providing illustrative examples to pinpoint specific weaknesses.
>
> 2. **Accuracy on newly tokenized inputs:** We are calculating the full accuracy for newly tokenized inputs in models with the original vocabulary to provide a comprehensive comparison.
>
> 3. **Estimated inference speed gain:**
> See general response.

---

> ### Comment · Area_Chair_a2kv · 2024-11-25
> **Please discuss further.**
>
> Have the authors clarified and answered your questions satisfactorily?

---

### Official Review · Reviewer_NB7M · 2024-11-02

**Soundness:** 3
**Presentation:** 2
**Contribution:** 3
**Rating:** 6
**Confidence:** 4

**Summary:**

In this paper, the authors investigate how LMs internally reconstruct word-level representations from sub-word tokens, a process they term "detokenization". They provide evidence that Lms can inherently combine sub-words into hidden representations which can be mapped into coherent words, even for out-of-vocabulary items, across the early to middle model layers. By probing LMs on both known words and artificial nonwords, they show that the model forms distinct representations for these categories, suggesting an "inner lexicon" that extends beyond tokenized inputs. The findings reveal that this detokenization mechanism leverages feedforward layers and attention patterns to generate whole word representations, which could, in theory, improve vocabulary flexibility without finetuning (though this is not shown in practice).

**Strengths:**

This paper addresses a crucial question: how can language models construct symbolic representations of entire words when their input comes from tokenizers that often fragment words in ways that disregard their morphological structure? Specifically, the authors investigate whether LMs internally form representations of morphological units that help bridge the gap between the tokenized input and the naturally holistic nature of words in language. Through experiments, the paper presents some evidence that whole-word representations emerge within the model’s hidden states, even when it processes fragmented word tokens. Additionally, the writing is clear, and the experiments are easy to replicate.

**Weaknesses:**

I believe there is a disparity between the paper’s claims and the experimental evidence provided to support them. Specifically, some of the experiments lend themselves to alternative interpretations, which could be clarified with additional baselines or experiments. The paper claims is that model come up with an "internal lexicon" that create hidden representations of "virtual" words, even when fed, e.g., word pieces as input. This is a claim on the computation carried out by the model, i.e., it is implied that there are some modules whose explicit computation is forming this internal lexicon. I am not sure that the experiments provide sufficient evidence for this claim:

* First, the "motivating experiment" in Section 3 lacks sufficient controls. The authors demonstrate that there is a linear separation in the hidden state between the representations of actual multi-token words and fictional ones created by randomly mixing word pieces. However, this separation could simply reflect the model's ability to distinguish between linguistically valid English morphology and nonconforming sequences, rather than providing evidence of "internal detokenization." For instance, an alternative hypothesis is that the model has learned distributional cues—such as suffixes like "ing" rarely appearing at the beginning of a word—which causes out-of-distribution effects in the hidden states when encountering atypical token sequences.

* In Section 4.1, the authors hypothesize that "if the model performs detokenization, it will represent the last token of a word similarly to the original word token." However, even if such similarity is observed, it could be attributed to the distributional properties of language rather than any explicit "detokenization" process. For instance, in the example provided in the paper where "cats" is split into "ca" and "ts," it is plausible that the pretraining corpus contains instances where this split occurs unnaturally, such as in URLs like "catsanddogs.com" (an actual website) or in cases with typos. Such occurrences might push the representation of "ca ts" closer to that of "cats" without requiring an explicit detokenization step. Furthermore, it is known that such similarities exists also in word2vec methods like Glove, and it is difficult to argue that any explicit detokenization happens there.

* In Section 4.2, the authors feed the hidden state of the last token of a multi-token word into the model and prompt it to repeat the word. Instances where the model accurately reproduces the entire word are taken as evidence that it has stored the multi-token word in an "internal lexicon." However, a key baseline is missing: including phrases that are not single words, such as "repeat this word: rainy day." The observed results could simply reflect the model's tendency to form contextualized representations that transfer information across tokens, rather than indicating an internalized whole-word representation.

* Finally, the paper’s closing sections aim to illuminate the model's internal computations and the supposed formation of an internal lexicon. While the results provide some evidence of contextualization in the feedforward layers, it's unclear to me whether they genuinely support the existence of an internal detokenization process. Intervention-based experiments could strengthen this claim. For example, could we identify a subset of parameters where ablation specifically impairs performance on multi-token words without affecting single-token words? Or could linear concept erasure techniques reveal a subspace whose neutralization removes all distinctions between multi-token and single-token representations?

**Questions:**

* The experiments in 4.1 focus on logit lens. What about cosine similarity or more direct measrues of similarity?

---

> ### Author Response · Authors · 2024-11-19
>
> Thank you for the constructive feedback. We are glad you found the paper clearly written and easily reproducible. We address specific comments below.
>
> **Improved vocabulary flexibility without finetuning has not been shown in practice** (from the summary)
>
> We would like to clarify that the practical application of vocabulary flexibility without finetuning is demonstrated in Section 6. There, we show how the detokenization mechanism allows for the generation of multi-token words not present in the tokenizer vocabulary, providing empirical evidence of the method’s effectiveness in practice.
>
> **Weakness 1**
>
> See general response.
>
> **Does token similarity stem from distributional properties in the pretraining data (section 4.1)?**
>
> Thank you for your valuable feedback. We agree that the distributional properties of the pretraining data are foundational to shaping the model’s inner lexicon, as seen in our observation that rare words (e.g., archaic terms) are retrieved less effectively than frequent ones (e.g., Wikipedia terms). However, our experiments suggest that retrieval from the inner lexicon is not merely an artifact of typos, artificial separations, or coincidental co-occurrences in the data.
>
> To address this, we tested artificial separations in words with general, multi-optional suffixes like **“ing”**, **“ion”**, and **“est”**. These suffixes lack strong ties to specific prefixes yet consistently resulted in high retrieval rates, suggesting that the proposed explanation (that the last token embeddings resemble the full word) is inaccurate. For instance, when **“running”** was split into **“runn”** and **“ing,”** the representation of “ing” closely matched the full word **“running,”** which cannot be explained by distributional properties alone. See https://imgur.com/GnsUuVw. We will revise the manuscript to clarify these points.
>
> **Information transfer or internalized whole-word representation?**
>
> Thank you for your insightful comment. The question of whether multi-word phrases are part of the model’s inner lexicon is indeed an important one. We hypothesize that some common multi-word expressions are also part of the internal lexicon. Properly studying this question is outside the scope of this study, as our focus here is specifically on single-word tokens.
>
> To address the core of your question—whether the model’s behavior reflects the information transfer across tokens or internalized whole-word representation, we ran a patching experiment similar to the one in the general response where we used the penultimate token representation in our word/nonword experiment. Particularly, we compared whether the model is able to reproduce the prefix (the full word excluding the last token) from the penultimate token. Our results  (https://imgur.com/ebIySuX) show that it does so far worse than the last word token, indicating that it has yet to build the word representation at this point. We stress that this is despite the high co-occurrence between that penultimate token and the previous sub-word token(s).
>
> **Intervention-based experiments to validate the internal detokenization process**
>
> Thank you for the suggestion regarding intervention-based experiments to validate the role of feedforward layers in detokenization. We conducted an ablation study on the word retrieval process, and the results strongly support the importance of FFNs in detokenization. We followed https://arxiv.org/pdf/2305.16130, and ablated the specific 5% of layers that hold the “memory” of a word (i.e., the layers from which we take the representation of the given word). Our results (https://imgur.com/AL4NcPS) show that this leads to the detokenization process failing entirely.
> In contrast, as a control, we ablated the same proportion of random FFN layers and observed almost no effect on the retrieval rate. This difference highlights the critical role of FFNs in forming and retrieving internal representations of words.
> We will incorporate these results into the manuscript.
>
>
> **Logit lens vs cosine similarity in Section 4.1**
>
> Thank you for your comment. We initially used cosine similarity to rank embeddings based on specific hidden representations. However, we transitioned to logit lens because results were similar, and logit lens offered a more streamlined framework for our experiments and analysis.
> Following your suggestion, we repeated the logit-lens artificial split experiments (Fig. 3a in the paper) with cosine similarity, and found that the results show very similar patterns (https://imgur.com/f6H42DT)

---

> > ### Comment · Reviewer_NB7M · 2024-11-21
> > **Response**
> >
> > Thanks for your response.
> >
> > 1. Distributional cues: first, I agree that "retrieval from the inner lexicon is not merely an artifact of typos, artificial separations, or coincidental co-occurrences in the data". I think, however, that together with contextualized word representations, this can explain (most) of the effect you see. In static embeddings models that give you a representations of OOV words like "runnning" you are probably going to see a similar effect, and these models compute a much simpler "contextualized" representations of OOV models than transformers. Your results can stem from the fact (1) "runn" is represented similarly to "run". (2) the contextualized representations of the last suffix token is influenced by preceding token.
> >
> > 2. Have you tried to replicate the "motivating experiment" with morphologically valid pseudo words?
> >
> > 3. Regarding the patching experiment: I think it shows that model create contextualized representations that focus the semantics of the word unit in the last token, which is known from many previous work. I don't see how your experiment differentiate the claims (1) models create contextualized representations [that happen to focus on the last token], and (2) models have an inner lexicon. Are these claims equivalent in your view?
> >
> > More fundamentally, I think the main problem with the paper at its current form is that it makes unsupported claims that do not stand scientific scrutiny. The paper claims that transformers come up with an "intrinsic lexicon" and that they "compute" words from tokens (the "computational" and "algorithmic" levels in [Marr's levels of analysis](https://en.wikipedia.org/wiki/David_Marr_(neuroscientist)#Levels_of_analysis)). These are algorithmic claims on the computation carried out in the transformer models, and I think they oversell the actual findings. I don't think the experimental setup in this paper valides, or even tests, these claims.

---

> ### Author Response · Authors · 2024-11-21
>
> Thank you for the quick and thoughtful response.
>
> We would first like to clarify what we mean by ‘inner lexicon’ and its relation to contextualization. When processing words split into multiple sub-word tokens, recent work identified a detokenization stage in the early layers of LMs [1,2,3], in which such tokens are converted into meaningful representations. In our work, we show evidence that to successfully perform detokenization and reconstruct the full word representation, the model needs to recall a “memory” of the word from its FFN layers–a process *beyond pure contextualization*. These layers were previously shown to emulate neural memories used to recall more abstract concepts [4,5]. We refer to this role of FFNs as the model’s ‘inner lexicon’. We can see how this term might seem abstract, and welcome any suggestions for a better term. Still, we find that this mechanism cannot be explained by contextualization alone.
>
> We note that we do not ignore the role of contextualization in the detokenization process: as we show in Section 5.2, attention plays a critical role in building the fused word representation. However, while static embedding models can build up word representations through contextualization alone, this does not warrant that Transformer models rely on a similar mechanism. Indeed, we show that the FFN layers complement contextualization and build the final representation. To further illustrate this, following your suggestion, we have presented in our previous response to you an ablation study that deletes the specific parts that represent the word from the FFN. As we have shown, doing this makes the model completely fail to recognize the word. This indicates that contextualization alone is insufficient for LLM detokenization.
>
> To further distinguish between hypotheses (1) and (2) in point 3 in your response, in our original general response we compared the *penultimate* token in multi-token words of length 3 or more against nonwords. That is, we took inner sub-word tokens that do not constitute a whole word (e.g., in the word ‘encyclopedia’, broken down into ‘en #cyc #lopedia’, we took the inner representation of the ‘#cyc’ token). Our results (https://imgur.com/OiTscK5) show that unlike the final tokens, penultimate tokens are poorly distinguishable from nonwords. In our response to you we also presented a similar experiment with patchscopes, also with the penultimate token, which showed a similar trend (https://imgur.com/ebIySuX) ---models cannot reproduce the prefix (e.g., ‘en #cyc’) from the penultimate token (‘#cyc’). This is in contrast to our main results, where taking the last token (e.g., ‘#lopedia’) allows the model to succeed in reproducing the full word. These results further support our claim that models rely on an ‘inner lexicon’ to build meaningful representations: when they fail to match a sequence of tokens with a word they can recall (e.g., as in the penultimate token), no meaningful representation is constructed.
>
> If hypothesis (1) was correct, you would have expected models to be able to distinguish such penultimate tokens from nonword tokens. What we show here is that the full-word representations do not only “*happen to focus on the last token*”, but rather that the last sub-word token is crucial for the model to match the contextualized representation with a word concept retrieved from memory and form a meaningful representation. We also note that while our focus is on words, we agree with your claim in the original review that it might also contain other units such as multi-word expressions. However, even if this is proven correct, they are still part of the inner lexicon we discuss, rather than being an artifact of contextualized representation.
>
> Finally, to your question (2), **yes**, in our original general response we also presented morphologically valid pseudo words---the ARC  Nonword Database, using only the subset of words that follow typical patterns in English spelling and morphology. Our results (https://imgur.com/OiTscK5) show a very similar trend to our original experiment, indicating that the separation we observed does **not** “*simply reflect the model's ability to distinguish between linguistically valid English morphology and nonconforming sequences*.”
>
>
> We hope our response has clarified our points. We are grateful for your feedback, which allowed us to further improve our paper and make our results much stronger.
>
>
> [1] Nelson Elhage et al., Softmax Linear Units, Transformer Circuits Thread, 2022.
>
> [2] Wes Gurnee et al., Finding Neurons in a Haystack: Case Studies with Sparse Probing, Trans. Mach. Learn. Res., 2023.
>
> [3] Sheridan Feucht et al., Token Erasure as a Footprint of Implicit Vocabulary Items in LLMs, 2024.
>
> [4] Mor Geva et al., Transformer Feed-Forward Layers Are Key-Value Memories, EMNLP 2021.
>
> [5] Kevin Meng et al., Locating and Editing Factual Associations in GPT, NeurIPS 2023.

---

> > ### Comment · Reviewer_NB7M · 2024-11-24
> > **Response**
> >
> > Thank you for extending the motivating experiment. This resolves all my concerns on the matter.
> >
> > I still don't manage to understand what you learn from the experiment on the penultimate token. I agree this convinces us that to the extent models combine information from tokens to create meaningful representations of whole words, this happens in the last token. You say: "the last sub-word token is crucial for the model to match the contextualized representation with a word concept retrieved from memory and form a meaningful representation". I can understand this as an hypothesis, but are you convinced that you provide evidence for this hypothesis?
> >
> > I'd appreciate it if you can elaborate on the ablation experiment you mention and what you learn from it. To my understanding, you show that you can isolate a small subset of the FFN parameters whose ablation prevents you from getting whole-word representations using logit lens. How does this tell you that the FFN layer is used as memory to retrieve from the inner lexicon? if this were the case, what effect would you expect to see for this ablation on the behavior (LM completions) of the model on (1) texts that are composed of multi-token words, and (2) texts that are composed of only single-token words? What do you observe in practice?

---

> > > ### Author Response · Authors · 2024-11-26
> > >
> > > Thank you for your thoughtful feedback and constructive suggestions.
> > >
> > > After much discussion among us, we think we might have an understanding of our different perspectives on our findings. When we say “lexicon” or “memory”, one might think of an explicit dictionary that maps each token to a single vector. We want to highlight that our intention, inspired by previous work on concepts in LLMs [1,2], is to a *soft* version of a lexicon, which both (a) combines multiple vectors to form concept representation; and (b) is not unique, that is—a word might be saved in memory in more than one layer. We agree that the term “lexicon” might be confusing in this sense, and are welcome to suggestions for alternative phrasing. With that in mind, we address your questions below.

---

> > > > ### Author Response · Authors · 2024-11-26
> > > >
> > > > Indeed, the experiment on the penultimate token, as you stated, shows that when the model combines information from sub-word tokens to form representations of whole words (i.e., detokenization), the meaningful representation emerges in the last token. But how does the model “know” it is currently processing the last sub-word token? We provide evidence for one explanation for this phenomenon, namely that models match the contextualized representation with a word concept retrieved from memory. We do so by studying the role of the FFN layers in detokenization.
> > > >
> > > > In our paper, we showed that the whole-word representations of words separated into multiple tokens using misspellings or artificial separations are retrieved from the FFN layers  *before* they emerge in the residual stream (Figure 4). Thanks to your suggestion, we also ran the ablation experiment (further discussed below). This experiment showed that zeroing out the FFN additions to the residual stream in the few (~5%) layers where they match the retrieved word, results in the model representation no longer matching the whole word in any of the layers.  While this result does not conclusively prove the existence of an internal lexicon, it aligns with prior findings [1,2] on the role of FFN layers as neural memories for storing abstract concepts. Altogether, this shows that models use FFNs to match the contextualized representation and retrieve the whole word from memory, and that contextualization alone is insufficient for forming a meaningful word representation (specifically shown in the ablation experiment).
> > > >
> > > > Next, we further elaborate on the ablation experiment from our initial response. Our methodology involves artificially splitting words into multiple tokens and evaluating whether the residual stream representation of the final token aligns with the original word embedding via logit lens. Ablating specific FFN layers identified as retrieving the word’s concept using logit lens—which results in *different* layers for each word—caused a sharp drop in reconstruction accuracy. Notably, if FFN layers merely enhanced contextualization without a memory function, one might expect a more gradual or uniform degradation across tokens. Instead, the sharp accuracy drop specific to these layers suggests a distinct role in retrieving pre-encoded representations. These findings support the hypothesis that the model uses a memory-like mechanism, queried after contextual aggregation, though the precise nature of this mechanism warrants further investigation.
> > > >
> > > > To your question about ablating single vs. multi-token words, we note that we do not hypothesize that the model’s memory is restricted to multi-token words only, but believe it contains both single- and multi-token ones. This is supported by our separation and typos experiments, which show that the model is able to reconstruct the original (single-token) word representation from different multi-token representations of that word. As a result, we do not expect a different behavior when applying ablation to both groups.
> > > >
> > > > To test this hypothesis, and to address your question on the effect of ablation on model behaviour, we ran the following experiment. We evaluated Llama-2-7B’s completions on the prompt “The capital of [COUNTRY] is ____” using all country names tokenized as a single token (taken from https://huggingface.co/datasets/akrishnan/country_capital_questions). We ran experiments with two types of conditions: (a) using the original (single-token) country name (e.g., “China”), vs. splitting it into multiple tokens (“Chi” + “na”); and (b) with and without ablation. For each case, we used the country name representation obtained by running logit lens on the FFN’s output before adding it to the residual stream. We evaluated the model on the proportion of correct capitals predicted as the next token (“Beijing”). Our results (https://imgur.com/4clzdbc) show a few interesting trends: First, comparing the single vs. multi-token word representation, we find a small degradation (89% vs 80%). However, when performing the ablation, both numbers go down substantially (to 72% and 52%, respectively), indicating that indeed both single- and multi-token word representations are stored in these FFN layers.
> > > >
> > > > To conclude, we stress that we do not claim to have fully mapped or described the model’s inner lexicon, but rather argue that modern LLMs use mechanisms beyond contextualization, with FFN layers storing and retrieving word representations. This is distinct from static embedding models, where contextualization alone suffices. Our findings highlight a more intricate process, complementing attention mechanisms.
> > > > We hope these clarifications address your concerns. Thank you again for your valuable feedback, which has helped us improve the work.
> > > >
> > > > [1] Mor Geva, et al. Transformer Feed-Forward Layers Are Key-Value Memories, EMNLP 2021.
> > > >
> > > > [2] Kevin Meng, et al. Locating and editing factual associations in GPT, NeurIPS 2022.

---

> > > > > ### Comment · Reviewer_NB7M · 2024-11-27
> > > > > **Response**
> > > > >
> > > > > I appreciate the engagement in your responses and have revised my score accordingly. I hope you decide to hedge your claims in the final version of this work. The field needs rigor rather than hype, and I believe your meaningful findings are undersold when presented in a hyped manner.

---

### Official Review · Reviewer_ChVq · 2024-11-02

**Soundness:** 3
**Presentation:** 2
**Contribution:** 3
**Rating:** 6
**Confidence:** 4

**Summary:**

This paper analyzes the process of latent detokenization inside the transformer-based LM forward pass, as it occurs across network layers. It shows that models are able to recognize words from pretraining even when they are noised with slight token variations or artificially split across multiple tokens. These experiments are different from earlier works, but ultimately show very similar findings about hierarchical processing in transformers. Using these findings, a novel method is briefly introduced to leverage the internal states of merged tokens to automatically expand the token vocabulary, which can hypothetically improve inference costs with fewer lookups. This method appears initially effective, but could be explored more.

**Strengths:**

1. This paper analyzes the process of detokenization across transformer network layers via a series of targeted experiments. It builds an intuitive understanding that agrees with many prior works in layer-based analysis.

2. The paper proposes an interesting method for training-free expansion of the model vocabulary by leveraging the insights into internal word representations. This method is shown to be effective in limited experiments. See below in "weaknesses" for further thoughts on this.

3. The writing is clear, but sometimes too abstract (see weakness 5).

This paper shows very solid work and I greatly appreciate the thorough breadth of exploration, though it could possibly be more effective to focus on fewer areas. I want to emphasize that I enjoyed reading the paper and believe it will be strong after some revision, including reworking the claims and focusing more on the novel contributions which are begun later in the paper. I believe it would be more impactful to explore sec 6 in more depth; see weakness 4 below.

**Weaknesses:**

1. The concept of an inner lexicon is interesting, but not novel as is claimed in this work. The idea follows implicitly from prior work in the memorization of training data, and explicitly in works about tokenization, such as the introduction of BPE (which is discussed greatly in this paper). It is the stated goal of subword tokenizers to enable learning a vocabulary of words and concepts which is larger than the vocabulary of concrete tokens through the process of token combination. It is nice to see these findings reproduced and analyzed, but they are not new.

2. The experiment in section 3, which motivates the idea of an inner lexicon, is not very strongly designed. Why are nonwords created by randomizing tokens, and not by some other method on the morphological level or otherwise something more linguistically motivated? Resulting nonwords do not seem to follow English conventional morphology (eg. the nonword "chha") and this could make it trivial to distinguish words from nonwords. Prior work has shown LLM sensitivity to word frequency in training corpora, and this experiment seems to reproduce those findings. This experiment seems to me to show that LLMs can distinguish easy cases such as "chha" which are very dissimilar to real words, and predictably struggles with more difficult cases that more closely resemble real words (see appendix) but there doesn't seem to be strong evidence that the LLM representation is doing more than locating words on a gradient based on their prior likelihood of appearing in the pretraining corpus. This fact is fairly well established at this point.

3. The experiments in the paper seem mostly sound and reasonable, but their novelty is overstated. Several of the earlier experiments in particular build on each other to show that early and intermediate layers in the network are responsible for aggregating and disambiguating word representations (sec 4 and 5). However, these findings may be seen to be subsumed by many prior works in the analysis of syntactic and semantic composition of tokens across transformer layers (see section 4 in [1] for many citations).

4. The paper may have been too ambitious in scope. The first several experiments were good reproductions of findings. The last experiment was novel to me, and it would have been interesting to expand on it more deeply. However, it did not require many of the earlier experiments in order to understand it, which took up most of the room in the paper. Other reviewers may have different opinions, but mine is that the paper would be more valuable if it explored the final research question more deeply, and provided more concrete findings for it. For example, can we estimate a size/contents of the inner lexicon? Does this lexicon scale with model capacity and/or training size? Can we provide some guarantees or estimates about the boundaries of the method of finetuning-free vocabulary expansion? For what kinds of words is this method effective and when is it ineffective?

5. There were many smaller experiments given in the paper, and this resulted in important implementation details being often omitted. For example, experiments often hinge on model memory of tokens from training, and the natural distributions of those tokens in the corpora, but details about how words/tokens were sampled in the tests (such as construction of nonwords) were not often given in enough detail to reproduce experiments. I would expect there to be significant influence of such distributions on test outcomes, so these details are important.


[1] Anna Rogers, Olga Kovaleva, Anna Rumshisky; A Primer in BERTology: What We Know About How BERT Works. Transactions of the Association for Computational Linguistics, 2020.

**Questions:**

1. How were tokens assembled into nonwords in sec 3? I am missing detail here which could be useful in understanding the method. I also do not understand what it means to "fit" a KNN classifier (which is non-parametric) -- were there representations used which were different from those taken from the model hidden states?
2. There was a claim made that the proposed method in section 6 can improve inference-time costs, though I cannot find any experiments or numbers for this in the paper. Can the authors point me to or provide any information about this? Thank you.

---

> ### Author Response · Authors · 2024-11-19
>
> We thank the reviewer for their thoughtful and constructive feedback. We are happy they found our work as introducing a *novel* and effective method, as being clearly written, and as introducing a breadth of experiments. We address specific concerns below.
>
> **Novelty of the inner lexicon concept** (weakness 1 and 3)
>
> We appreciate the reviewer’s feedback and the opportunity to clarify. While prior work explored tokenization (e.g., BPE) and subword vocabularies, as well as word disambiguation, our focus is on how subword tokens are processed and aggregated into word representations. Our study shows that an internal lexicon of words exists, and demonstrates the **full detokenization mechanism**, involving both attention and feed-forward layers, culminating in the final layer’s coherent word representation.
>
> We appreciate the reviewer’s reference to the BERTology survey paper. The most relevant work we found there was the study on BPE attention heads in BERT [1], which showed that some of the attention heads attend to the previous sub-word tokens of the current token. We will add this reference.
>
> However, we are not aware of any paper that explicitly defines the **full process of detokenization**—specifically, how both attention and feed-forward layers contribute to aggregating the meaning of the full word into the last token representation, as we demonstrate in decoder-only models. We are also not aware of experiments showing that models can distinguish between words and nonwords, experiments that show that this representation is robust to typos, and that these representations can be fed into the model as input and be “understood” by it as we have shown. If there are other papers we missed that demonstrate these phenomena, we would greatly appreciate their references.
>
> **The experiments in section 3**
>
> See general response.
>
> **Expansion of the final experiment (section 6)**
>
> We fully agree that expanding the final experiment would enhance the paper. We address your specific suggestions as follows:
> 1. **Scaling with model capacity/training size:** This is an excellent question. We are currently conducting experiments with larger models to assess how our results scale with capacity and training size. We will share our findings as they become available.
> 2. **Effectiveness of different word types:** We are currently running experiments to identify the kinds of words for which this method is effective or ineffective. We will share our findings as we have them.
> 3. **Inner lexicon size/contents:** This question is very interesting, and we are currently exploring it. We note though that it is also quite challenging. For instance, it is not clear what to do with morphological inflections (e.g., plural, gerund, etc.). Does the model hold a different representation for each inflected form? Is there another mechanism for representing them? Another key question is the role of the word frequency in the pre-training corpus. As such corpora are typically unavailable, even extracting these frequencies is non-trivial. A simpler approach we are considering is starting with only base forms, by iterating a large dictionary while applying our method. We welcome other suggestions on how to tackle this problem!
> 4. **Boundaries of finetuning-free vocabulary expansion:** Another great question! As mentioned above, we believe pre-training frequency is a key factor here, and are currently exploring it. It would be great to come up with a scaling law of word frequencies that predict whether or not they are part of the inner lexicon.
>
> **Missing implementation details**
>
> We provide implementation details about the construction of nonwords in the general response. We note that our results show that despite the reviewer’s concern, our method is quite robust to different methods of generating nonwords. If there are other parts of our paper where implementation details seem insufficient, we would appreciate specific feedback on how to address these gaps. We will revise the manuscript to include a more detailed description of the dataset creation process and clarify any missing details across other experiments.
>
> **"Fitting" a KNN classifier**
>
> Thank you for pointing out the incorrect terminology. You are correct that “fit” is not the appropriate term for describing the k-nearest neighbors (KNN) process. We will omit it from the manuscript. To further clarify, the representations used in our experiments were only the model hidden states.
>
> **Improving inference-time costs**
>
> See general response.
>
> [1] Adaptively Sparse Transformers (Correia et al., EMNLP-IJCNLP 2019)

---

> ### Author Response · Authors · 2024-11-22
> **Scaling with model capacity/training size**
>
> As noted in our response to the question on scaling with model capacity/training size, we ran experiments with larger models (Llama2-13B vs. Llama2-7B). Patchscopes results improved from 77.4% to 85%, demonstrating that larger models may better leverage inner lexicons.

---

> ### Comment · Reviewer_ChVq · 2024-11-24
> **Raising score**
>
> Thank you to the authors for updating us with new results and a detailed discussion of weaknesses. My issues with the "motivating" experiment (sec 3) have been addressed by the added baseline, which was convincing. I also appreciate the authors' effort in addressing the concerns of reviewer NB7M -- I agree with many of the reviewer's points, but I am at least convinced of the soundness of this work as is, after the rebuttal. I will raise my score to 6 to reflect this.
>
> I do still think the authors' claims of novelty may be a bit inflated. Many investigations on how representations are combined in transformer network layers were done in the last few years, especially around the time BERT was popular (see for example [1]). I was not surprised to read some of the findings in this paper in the earlier sections, such as that middle layers were most activated; end-word token representations are the most summative (reminds me of e.g., the CLS token in BERT, though the nature of generative decoding in the models studied in this context may be an additional factor here); etc. However, there is value in these focused experiments and novelty in the later experiments.
>
> [1] Assessing Phrasal Representation and Composition in Transformers (Yu & Ettinger, EMNLP 2020)

---

### Official Review · Reviewer_rVMS · 2024-11-04

**Soundness:** 4
**Presentation:** 4
**Contribution:** 3
**Rating:** 8
**Confidence:** 4

**Summary:**

This paper explores the process in which models transform tokens, which often split long words into subwords (e.g., "un" "h" "appiness"), into higher level representations for the full word through "detokenization". Detokenization has been observed in LMs before, but has not been directly studied extensively. This work shows that LMs can recognize when a word is part of a larger word, and show that early attention fuses subwords together (in the last token of the word), and uses early MLP layers to then recall the full word from multiple subwords in an "internal dictionary" (e.g., representing "unhappiness" as a single vector internally even though it is not in the tokenizer). The authors then show that this can be used to expand a model's tokenizer by including the hidden 'internal dictionary' representation as an input token. This works to some extent.

Overall, this paper enhances our understanding of early layer processing in language models, and provides a path towards enhancing models to reduce inference time.

**Strengths:**

This paper answers particular unanswered questions surrounding "detokenization", which has been repeatedly observed and discussed without being properly studied. These are important for observations around, for example, stages of inference in language models.

Interpretability results on early layers of LMs are often lacking, as vocab projections are much easier to perform at later layers. This work provides interesting and convincing results for one role early layers take on in these models, which is indeed different from the roles of later layers.

The vocab expansion experiments are a nice proof of concept, and could be expanded on in the future to decrease inference times.

The results on typos are interesting and to my knowledge, novel

**Weaknesses:**

The evidence for a third stage of processing in Figure 2b is a little sparse. These results are only for one model, and the degree to which accuracy drops is not substantial enough to obviously be due to a difference in processing altogether. These results could be made stronger by including results for more models. As a motivating example, it is fine, but perhaps isn't the best use of that space if this point can't be made more strongly.

Typos:

L370: "form"

**Questions:**

If the model wants to generate some multitoken word that it represents in its 'internal dictionary' is it "planning" multiple tokens ahead? Why or why not?

---

> ### Author Response · Authors · 2024-11-19
>
> We are grateful for the constructive feedback, and are happy that the reviewer found our results as enhancing our understanding of early layer processing in LLMs and providing a path towards reducing inference time, our work as addressing key unanswered questions around detokenization, and our results as interesting, *convincing*, and *novel*. We address specific concerns below.
>
> **Evidence for a third processing stage in Fig. 2b is limited**
>
> Thank you for your feedback. While this is our primary focus, our results consistently show a drop around the middle of the network across all graphs and metrics (e.g., Figures 3 and 4 in the paper), indicating a potentially distinct phenomenon. Following your suggestion, we reproduced the experiments in Fig. 2b with three other models (Llama3-8B, Mistral-7B, and Yi-6B). We found that the same pattern across all models. See https://imgur.com/mlnJPcn.
>
> **Planning multiple tokens**
>
> Thank you for raising this insightful question. Your observation aligns with our intuition and presents a natural extension of this work. While we focus on the early stages of “detokenization,” it seems plausible the model plans multitoken words internally, adapting the representation to the first token if the word is out of vocabulary. Interestingly, our results in Section 6, which show that the fused word representation is the top-1 predicted token in 20% of the cases, indicates a potential mechanism for this planning: the fused word representation might be similar to the first sub-word token. As a result, the model is trying to predict the full word, but if it doesn’t exist, the first sub-word token is the next most similar token, and is thus selected as the next one. Continuing to explore this question is an exciting direction for future work.

---

> > ### Comment · Reviewer_rVMS · 2024-11-24
> > **Thanks for the new experiments and controls**
> >
> > Thank you for the response. I appreciate the additional models being tested for Fig 2b. and I think it helps strengthen the point.
> > I've also read and understand the feedback from other reviewers, and a few good points are raised. In particular, the need for more controls for word vs. nonword experiments **which I believe the authors properly address in their rebuttal.**
> >
> > I disagree with concerns over novelty. "Detokenization" is a somewhat broadly mentioned phenomenon in interpretability literature, but has not been properly studied. This paper takes a solid approach to studying it and reports some interesting results. I agree with ChVq that the last question could be investigated more deeply in the camera ready (the authors seem to have made progress with this during the discussion), so I maintain my score.

---

### Author Response · Authors · 2024-11-19

We thank the reviewers for their thoughtful and constructive feedback. We are encouraged that they recognized our paper as enhancing our understanding of early layer processing in LLMs and providing a path towards reducing inference time (rVMS), as addressing key unanswered questions around detokenization (rVMS), as providing interesting, *convincing*, and *novel* results (rVMS), as introducing a *novel* and effective method (ChVq), as being clearly written (ChVq, NB7M, t1G5), for the breadth of our experiments (ChVq), and for being easily reproducible (NB7M).

For you convenience, we reiterate our main contributions below:

1. We establish that LLMs hold an internal lexicon of whole-words, which goes beyond the tokenizer’s scope.
2. We show that this lexicon is robust to non-morphemic splits, typos and to out-of-vocabulary words.
3. We show that LLMs can “understand” internal representations of words in this lexicon in a *training-free* manner: when feeding the inner representation of such words to the model as input vectors, it can “understand” them despite never seeing them during training.
4. Moreover, we also show that the LLMs can generate such vectors, also without additional training, despite never seeing such vectors as neither input nor output.
5. We present initial results that these findings can help reduce both input and output sequence length, thereby potentially reducing both LLM cache size and decoding time.
6. We present evidence of the different components of the detokenization process, linking it to both the attention and feedforward mechanisms.

We recognize the reviewers’ suggestions to refine our claims, emphasize the novel aspects of our work, and expand specific sections, particularly Section 6. We addressed many of their suggestions. We list the major revisions we made below, and provide specific details in the individual responses:

1. We added two additional controls for our word vs. nonword experiments, which further support our main claim: models are able to internally distinguish between words and nonwords (ChVq, NB7M).
2. We repeated the experiment in Fig. 2b with three other models, and observed a very similar trend (rVMS).
3. We analyzed the retrieval rates of common word suffixes (e.g., “ing”), showing that their representations can be used to retrieve the full word (e.g., the word runn**ing**), thereby indicating that our observed phenomena are not simply due to token co-occurrence in the pretraining corpus (NB7M).
4. We added a patching experiment to compare the model’s ability to reproduce full words versus nonwords and prefixes (penultimate token representations) in the word/nonword setup. The model performed significantly worse with prefixes and nonwords, demonstrating that the full word representation is uniquely formed for complete words at the final stage, despite high token co-occurrence in prefixes (NB7M).
5. We added ablation experiments that drop FFN layers that build the fused word representation. We observed that this leads to the detokenization process failing altogether, which further supports the role of the FFN component in this process (NB7M).
6. We repeated the logit-lens experiments with cosine similarity, and observed a very similar trend (NB7M).


Finally, we address two of the comments that repeated among reviewers.

---

> ### Author Response · Authors · 2024-11-19
> **Regarding the construction of the nonword dataset in Section 3 (ChVq, NB7M)**
>
> We first highlight that the nonword dataset was constructed to address two potential artifacts in token representation: (1) tokens that commonly appear in specific positions (e.g., “ing” as a suffix or capital letters at the beginning of words), and (2) tokens more frequently used in real words. To mitigate these biases, we shuffled tokens from a 30,000-word dataset sourced from the Gutenberg corpus, grouping tokens by their positional indices and sampling tokens from each group to create nonwords. This process ensured that tokens retained their natural positional usage, and that biases like the ones raised by NB7M (nonwords with “ing” as the first token) were avoided. See https://imgur.com/LkCRLCo for a slightly updated version of Fig. 2a in the paper that illustrates this process.
>
> Following both reviewers' concerns, we further tested the robustness of our results by conducting two additional experiments.
> 1. To alleviate the concern that our nonwords do not seem to follow English conventional morphology, we conducted an additional words vs. nonwords experiment, this time using a dataset of linguistically plausible nonwords designed to resemble real English words [1]. Interestingly, the model performed better on this dataset, showing a stronger ability to distinguish between real words and these plausible nonwords compared to our artificially constructed nonwords.
> 2. Another concern raised by both reviewers was that our word vs nonword experiment was an artifact of word co-occurrence. To test this hypothesis, we conducted another experiment, this time comparing the pen-ultimate token against nonword tokens in words of length 3 or more. That is, we take a token that frequently co-occurs with the previous sub-word tokens, but does not constitute a whole word (e.g., in the ‘un-h-appiness’ example in fig1 in the paper, we took the inner representation of the ‘h’ token). Such tokens are naturally as frequently co-occurred with their prefix as their final token is (‘appiness’). Our results show that unlike the last tokens, the pen-ultimate tokens are poorly distinguishable from nonwords. See https://imgur.com/OiTscK5 for both results.
>
> Combined, our results suggest that the model’s ability to separate words from nonwords is neither due to the gradient of prior likelihoods from the training corpus, nor to morphological differences. Instead, they reflect more nuanced patterns in its internal representations of recognizing whole words. We will revise the manuscript to clarify the rationale behind the dataset construction, and include details of these new experiments.
>
> [1] Rastle, K., Harrington, J., & Coltheart, M. (2002). 358,534 nonwords: The ARC Nonword Database. Quarterly Journal of Experimental Psychology, 55A, 1339-1362

---

> > ### Author Response · Authors · 2024-11-19
> > **improving inference-time costs (NB7M, t1G5)**
> >
> > The potential for inference-time cost reduction is massive. Our experiments show that our models recognize ~70% of the multi-token words when given as input. Further, they use the fused word representation as their top-1 option in 20% of the cases when generating output. Importantly, this is all without any fine-tuning; we expect to see much higher numbers if we allow the models to adapt to using these representations.
> >
> > However, we note that the cost reduction also relies on the dataset we work with, and particularly the token-to-word ratio. As reducing costs was not our main motivation, we used the (English) wikitext2 dataset, which has a ratio of only 1.15 (i.e., there are about 15% more tokens than words). In our experiments, this leads to a relatively small potential of reducing costs, at most 11% of the average input sequence length, and 3% of the output sequence length. We also note that as input is processed in parallel, we do not expect much reduction in running time, but rather a reduction in KV cache requirements.
> >
> > We also note that we are currently working on experiments in other languages, which have a much higher token/word ratio (e.g., for GPT-2, this ratio is 4.17 for Arabic, [2]), and thus a much higher potential for cost reduction.
> >
> > [2] Sengupta et al., 2023. Jais and Jais-chat: Arabic-Centric Foundation and Instruction-Tuned Open Generative Large Language Models. https://arxiv.org/abs/2308.16149

---

### Meta-Review · Area_Chair_a2kv · 2024-12-09

**Metareview:**

This paper describes the process by which language models "detokenize" subword-level tokenization. Based on the observation that this process is robust to the addition of out of vocabulary words, they also propose a method for expanding model vocabulary.

Pros: Convincing experiments.

Cons: Some findings, and even the experiments accompanying them, are already made in the existing background literature, even from years ago. (Specifically the detokenization process involving the middle layers of the model.) The authors need an extended literature review in response to these concerns about lack of novelty. Tone of the paper exaggerates the result by describing their findings as the model's "internal lexicon" and is unjustifiably specific about the intuition of the process.

**Additional Comments On Reviewer Discussion:**

Authors ran extensive experiments in response to reviewer criticisms, and all negative reviews responded by raising scores. Reviewer rVMS disagreed with ChVq, who cited a lack of novelty and prior work extending back several years. The initially skeptical reviewers have been satisfied as to the experiments, but maintain two criticisms: ChVq believes that the literature review is insufficient, and NB7M argues that the specific framing and algorithmic intuitions are stronger than justified. Both of these criticisms are valid and I hope that the authors revise their paper to qualify their framing and to expand their literature review, reflecting which of their claims restate observations in previous models.

---

### Decision · Program_Chairs · 2025-01-22

Accept (Poster)